# Mitochondrial Ca$^{2+}$ uptake by the MCU facilitates pyramidal neuron excitability and metabolism during action potential firing

Christopher J. Groten [1✉] & Brian A. MacVicar[1✉]

Neuronal activation is fundamental to information processing by the brain and requires mitochondrial energy metabolism. Mitochondrial Ca$^{2+}$ uptake by the mitochondrial Ca$^{2+}$ uniporter (MCU) has long been implicated in the control of energy metabolism and intracellular Ca$^{2+}$ signalling, but its importance to neuronal function in the brain remains unclear. Here, we used in situ electrophysiology and two-photon imaging of mitochondrial Ca$^{2+}$, cytosolic Ca$^{2+}$, and NAD(P)H to test the relevance of MCU activation to pyramidal neuron Ca$^{2+}$ signalling and energy metabolism during action potential firing. We demonstrate that mitochondrial Ca$^{2+}$ uptake by the MCU is tuned to enhanced firing rate and the strength of this relationship varied between neurons of discrete brain regions. MCU activation promoted electron transport chain activity and chemical reduction of NAD$^+$ to NADH. Moreover, Ca$^{2+}$ buffering by mitochondria attenuated cytosolic Ca$^{2+}$ signals and thereby reduced the coupling between activity and the slow afterhyperpolarization, a ubiquitous regulator of excitability. Collectively, we demonstrate that the MCU is engaged by accelerated spike frequency to facilitate neuronal activity through simultaneous control of energy metabolism and excitability. As such, the MCU is situated to promote brain functions associated with high frequency signalling and may represent a target for controlling excessive neuronal activity.

[1] Djavad Mowafaghian Centre for Brain Health, University of British Columbia, Vancouver V6T 1Z3, Canada. ✉email: chris.groten@ubc.ca; bmacvicar@brain.ubc.ca

The activation of neurons and neuronal networks is fundamental to brain function and is critically dependent on ATP production by mitochondria[1–4]. In addition, these organelles influence reactive oxygen species production, apoptotic cascades, metabolite generation, and $Ca^{2+}$ signalling[5–9]. Of particular interest to the relationship between neuronal activity and mitochondrial function is the uptake of cytosolic $Ca^{2+}$ by the mitochondria. This process relies on the inner mitochondrial membrane potential and is mediated by the mitochondrial $Ca^{2+}$ uniporter (MCU). The MCU is a $Ca^{2+}$-selective ion channel that is normally closed, but opens in response to elevations of cytosolic $Ca^{2+}$[10–12]. The $Ca^{2+}$-dependent activation of the MCU is mediated by the regulatory subunits MICU1 and MICU2, which associate with the channel[11,13]. Physiological elevation of mitochondrial $Ca^{2+}$ via the MCU can enhance oxidative phosphorylation by facilitating the activity of the ATP synthase and dehydrogenases in the tricarboxylic acid cycle (TCA)[8,14–17]. The uptake of $Ca^{2+}$ by the mitochondria can also shape the magnitude and spatio-temporal dynamics of cytosolic $Ca^{2+}$ and thereby control $Ca^{2+}$-dependent signalling[9,18–21]. In synaptic terminals, these effects of the MCU control synaptic transmission and promote local ATP production[2,22–24]. Aside from its physiological role, excessive mitochondrial $Ca^{2+}$ uptake can disrupt organelle function and may contribute to neuronal death during acute excitotoxicity and progressive degenerative disorders, such as Alzheimer's and Parkinson's disease[25–29]. This highlights the importance of examining the mechanisms which govern mitochondrial $Ca^{2+}$ uptake and the associated implications for neuronal function in the brain.

MCU activation may have a critical role in governing information processing in the brain. Numerous studies have demonstrated functions for mitochondrial $Ca^{2+}$ uptake in neurons. However, these experiments were often performed under pathological conditions and in cultured cells[2,27,30–33]. Moreover, previous measurements of mitochondrial $Ca^{2+}$ in brain tissue were performed using $Ca^{2+}$-sensitive dyes which do not selectively localize to the mitochondria and can have adverse effects[32,34,35]. Consequently, the functional relevance of mitochondrial $Ca^{2+}$ uptake to neurons during information processing in the mammalian brain is unclear. Interestingly, it was recently shown that the MCU in excitatory neurons is required for sustaining fast neuronal network oscillations in rodent hippocampal tissue[36]. Moreover, in vivo imaging with a genetically encoded mitochondrial $Ca^{2+}$ sensor revealed that sensory or motor evoked brain activation triggered robust mitochondrial $Ca^{2+}$ elevations in the somatodendritic compartment of excitatory pyramidal neurons[37]. These studies point to a potentially important role for the MCU and mitochondrial $Ca^{2+}$ signalling during heightened brain activity, but several outstanding issues remain unaddressed. For example, the relationship between single neuron activity and mitochondrial $Ca^{2+}$ uptake has not been established. This may be important considering that signalling pattern determines metabolic demand, synaptic transmission and plasticity[1,38–40]. Additionally, discrete cell types and brain regions can exhibit distinct metabolic profiles, $Ca^{2+}$ channels, and mitochondrial properties[41–44], indicating that the relevance of mitochondrial $Ca^{2+}$ uptake to specific neuronal populations may vary. Lastly, the functional importance of mitochondrial $Ca^{2+}$ uptake in the somatodendritic compartment of excitatory neurons during brain activity is not certain, but it could influence numerous processes, including adaptive energy metabolism, excitability, synaptic plasticity, and gene expression[21,45–47].

In the present study, we examined the functional relevance of the MCU during neuronal activation by determining the relationship between action potential firing and mitochondrial $Ca^{2+}$ uptake in excitatory neurons of the cortex and hippocampus.

Using a genetically encoded mitochondrial $Ca^{2+}$ sensor, we demonstrate that the MCU activates in response to increasing action potential firing rate to initiate long-lasting mitochondrial $Ca^{2+}$ elevations in the somatodendritic compartment of pyramidal neurons. By doing so, the MCU governed activity-dependent changes in mitochondrial energy metabolism and reduced the activation of the slow afterhyperpolarization (sAHP), which mediates negative feedback control of excitability. These results suggest that the MCU is engaged by changes in spike firing frequency to facilitate signal processing in pyramidal neurons of the brain.

## Results

**In situ characterization of cytosolic and mitochondrial $Ca^{2+}$ in pyramidal neurons during action potential firing.** In order to measure mitochondrial $Ca^{2+}$ in pyramidal neurons, we drove the expression of mitoRGECO1.0 (mRGECO) in the rat brain by intracortical injection of adenoassociated virus (AAV) (Supplementary Fig. 1)[48,49]. This sensor is a genetically encoded $Ca^{2+}$-sensitive fluorescent protein that localizes to the mitochondria due to the presence of the targeting sequence from cytochrome c oxidase subunit VIII[48]. mRGECO expression was controlled by the synapsin promoter to achieve its neuronal localization. This sensor was chosen because it is sensitive to modest $Ca^{2+}$ concentrations ($K_d = 0.48\,\mu M$) and has red emission, which allows for it to be imaged simultaneously with green fluorescent sensors[48]. Following AAV injections, two-photon imaging of acute cortical and hippocampal brain slices showed the mitochondrial localization of mRGECO in the neuropil and pyramidal neuron cell bodies (Supplementary Fig. 1). In unstimulated cortical brain slices, mRGECO fluorescence was largely stable in the soma and neuropil over time (Fig. 1a, b). On occasion, there were local transient increases in mRGECO fluorescence (peak $\Delta F/F_o = 0.18 \pm 0.026$, n = 4 slices, N = 3) which presumably occurred as a result of spontaneous neuronal signalling (Fig. 1a, b). To monitor mitochondrial $Ca^{2+}$ dynamics in relation to evoked neuronal activity, we performed whole-cell patch-clamp recordings from layer 3-5 pyramidal neurons expressing mRGECO. Cells were filled with Alexa-488 and confirmed to be pyramidal neurons based on the presence of a pyramid-shaped cell body, large apical dendrite, and multiple basal dendrites (Fig. 1c). Neuronal excitation was elicited using a 50 Hz, 4-sec train of depolarizing current pulses (+2 nA, 5 ms each), with each pulse consistently evoking a single action potential (Fig. 1c). Shortly after the onset of action potential firing, a punctate pattern of increased mRGECO fluorescence became apparent in both the soma and dendrites of pyramidal neurons, indicating that strong spike firing was sufficient to increase mitochondrial $Ca^{2+}$ (Fig. 1c–e). The elevation of mitochondrial $Ca^{2+}$ was very long-lasting and recovered to pre-stimulus baseline in a monoexponential fashion over several minutes (somatic $\tau = 145.5 \pm 11.15$ sec, $n = 7$, N = 4) (Fig. 1d). This protracted recovery of $Ca^{2+}$ is consistent with known properties of mitochondrial $Ca^{2+}$ dynamics and is mediated by the mitochondrial $Na^+/Ca^{2+}$ exchanger[48,50]. Regional comparison of evoked responses showed that the peak mRGECO fluorescence change was greater in the soma relative to the apical dendrite (Fig. 1e). This may reflect differences in $Ca^{2+}$ influx[51] or the fraction of cell volume occupied by the mitochondria between the soma and dendrites.

$Ca^{2+}$ uptake into the mitochondria occurs primarily via the MCU, although other $Ca^{2+}$ entry routes have been suggested[52–55]. To examine this, we tested the effect of Ru360, an MCU inhibitor[56], on mitochondrial $Ca^{2+}$ uptake during spike firing. As Ru360 has limited membrane permeability[57], it was

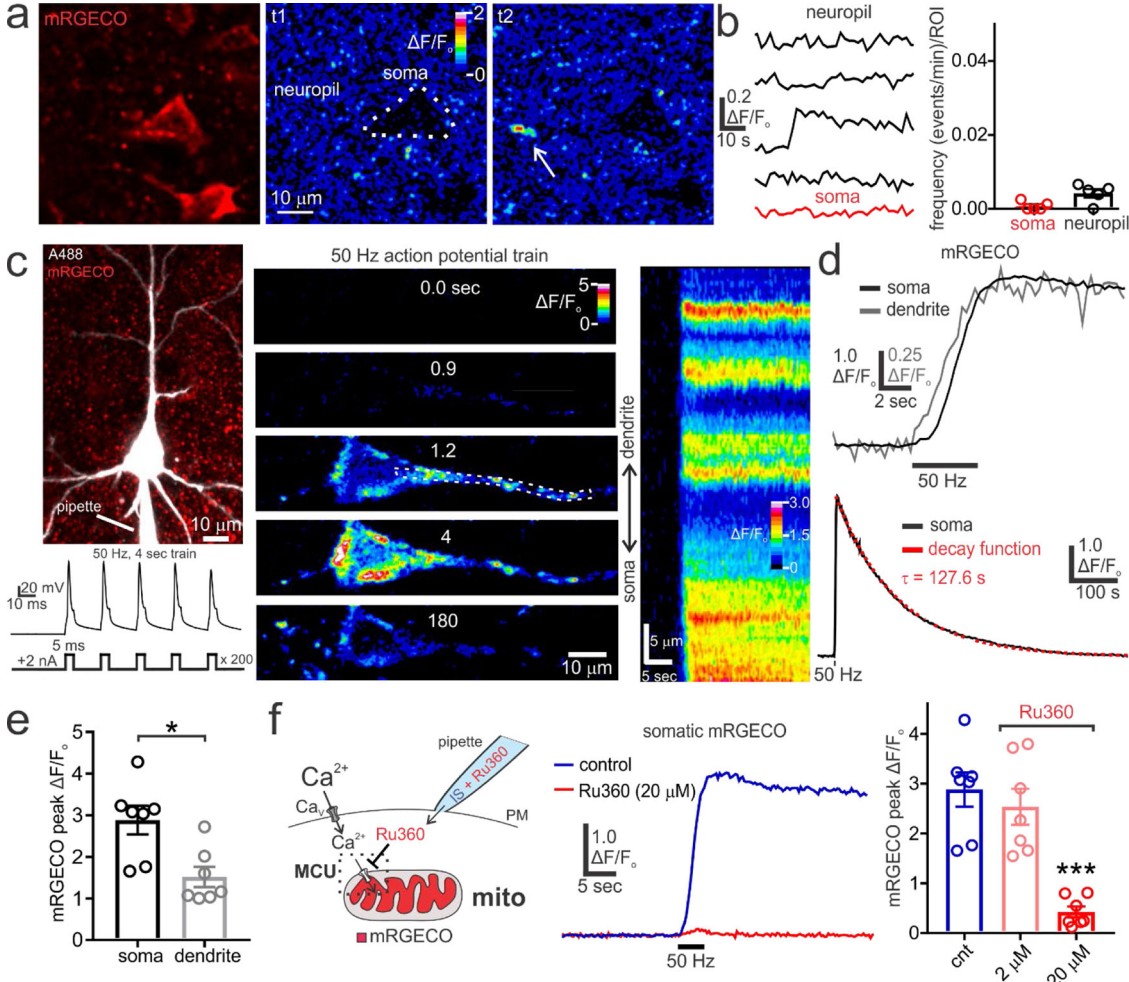

**Fig. 1 Pyramidal neuron action potential firing triggers a long-lasting mitochondrial Ca²⁺ elevation mediated by the MCU. a** mRGECO fluorescence in neurons of the cortex is stable over time ($\Delta F/F_o$) (time 1 [t1] vs time 2 [t2]), apart from infrequent spontaneous fluorescence transients seen in the neuropil (*arrow*). **b** *Left*, mRGECO fluorescence from regions of interest (ROI) in the soma and neuropil, showing the stability of the signal and an occasional spontaneous event. *Right*, Summary data demonstrates the low rate of spontaneous mitochondrial Ca²⁺ elevations, which did not significantly differ between the neuropil and the soma (soma: $n = 5$, $N = 3$; neuropil: $n = 5$, $N = 3$; Mann-Whitney test, $p = 0.063$). **c** *Upper Left*, Maximum intensity projection of a patch-clamped pyramidal neuron expressing mRGECO and dialyzed with Alexa488 (A488). A 50-Hz train of depolarizing current pulses (+2 nA, 5 ms each) elicited under current-clamp recording conditions (*Lower Left*) produced a marked increase in mRGECO fluorescence ($\Delta F/F_o$) in the cell body and apical dendrite that lasted for several minutes (*Middle & Right*). Kymograph reveals the spatial pattern of mitochondrial Ca²⁺ uptake along the somatodendritic axis after the onset of the action potential train. **d** *Upper*, mRGECO fluorescence measured from ROIs in the soma and apical dendrite shows the onset of mitochondrial Ca²⁺ loading shortly after the start of the action potential train. *Lower*, The somatic mitochondrial Ca²⁺ levels slowly recovered to prestimulus baseline in a monoexponential manner (decay time constant [$\tau$]). **e** Summary data showing a significantly larger evoked mRGECO fluorescence change in the soma relative to the apical dendrite (soma: $n = 7$, $N = 4$; dendrite: $n = 7$, $N = 4$; paired *t* test, $p = 0.031$). **f** *Left*, Illustration depicting the method for blocking the MCU by intracellular dialysis of Ru360 via the internal solution (IS) of the patch pipette. *Middle & Right*, Action potential evoked mitochondrial Ca²⁺ uptake is significantly reduced by 20 μM Ru360, but not 2 μM Ru360, relative to control (control: $n = 7$, $N = 4$; 2 μM Ru360: $n = 7$, $N = 2$; 20 μM Ru360: $n = 7$, $N = 3$; One-way ANOVA with Dunnett's multiple comparisons test: 2 μM Ru360, $p = 0.62$; 20 μM Ru360, $p < 0.0001$). The Control data set is reproduced from the summary data in panel E. Summary data presented as the mean ± SEM. *$P < 0.05$, ***$P < 0.001$. The number of cell replicates is shown in the graphs as well as the 'n' value in the text. The number of animal replicates is represented by the 'N' value. Source data used for all summary figures are provided in the Supplementary Data 1 file.

dissolved in the internal solution of the pipette and delivered to neurons via intracellular dialysis (Fig. 1f). Ru360 at 20 μM, but not at 2 μM, reduced the magnitude of mRGECO fluorescence changes evoked by action potentials (Fig. 1f). Although a small residual mitochondrial Ca²⁺ elevation was usually present at this concentration of Ru360, these data confirm that the MCU is the principal route for mitochondrial Ca²⁺ uptake in pyramidal neurons during action potential firing.

Cytosolic Ca²⁺ derived from either Ca²⁺ entry across the plasma membrane or Ca²⁺ release from the endoplasmic reticulum can activate the MCU and increase mitochondrial Ca²⁺[9,10,58]. We therefore, examined the relationship between cytosolic Ca²⁺ and mitochondrial Ca²⁺ levels during action potential firing. Removal of extracellular Ca²⁺ from the ACSF completely eliminated spike-evoked mitochondrial Ca²⁺ uptake (Fig. 2a), suggesting that the source of cytosolic Ca²⁺ driving the mitochondrial Ca²⁺ elevation is plasma membrane Ca²⁺ influx, rather than intracellular Ca²⁺ release. This Ca²⁺ influx is likely mediated by the activation of voltage-gated Ca²⁺ channels in the soma, as well as the dendrites through backpropagating action

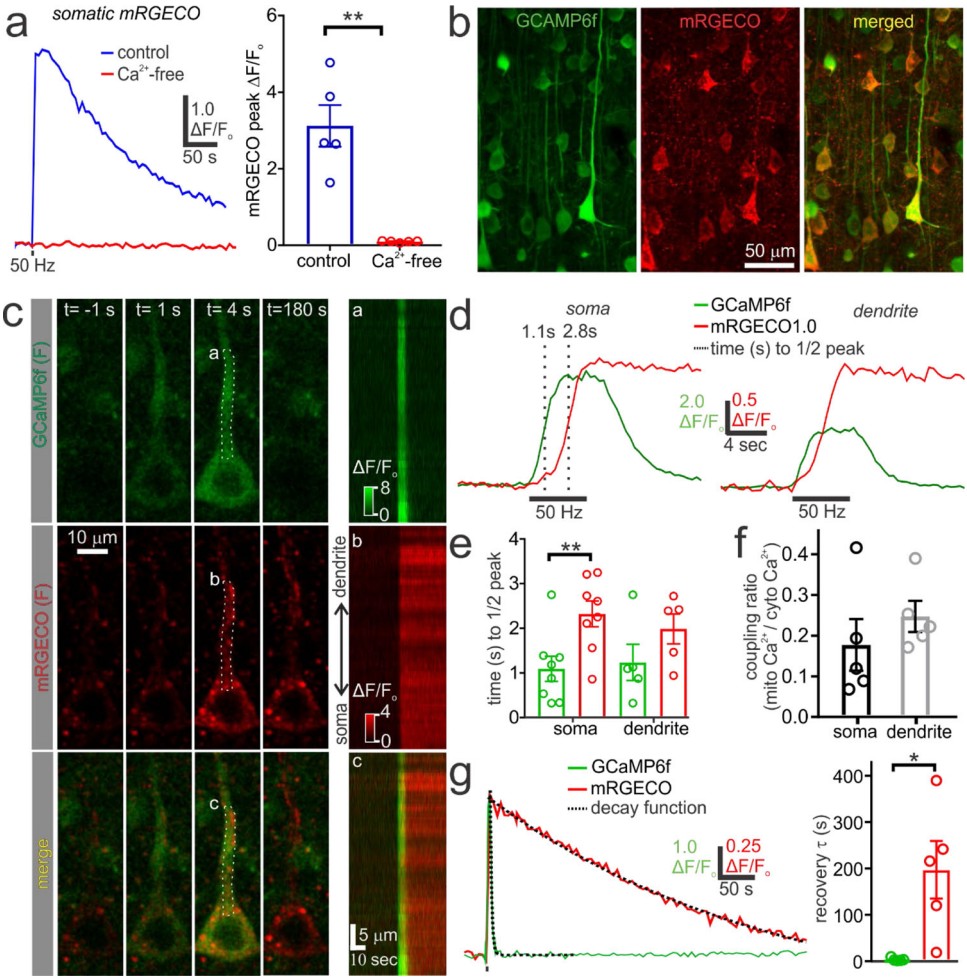

**Fig. 2 Simultaneous imaging of GCaMP6f and mRGECO reveals the distinct dynamics of cytosolic and mitochondrial Ca$^{2+}$ during action potential firing. a** *Left*, Somatic mRGECO fluorescence changes from cortical pyramidal neurons during a 50-Hz, 4-sec action potential train in the presence (control) and absence of extracellular Ca$^{2+}$ (Ca$^{2+}$-free). *Right*, Summary data shows a significant reduction in evoked mitochondrial Ca$^{2+}$ uptake in the absence of extracellular Ca$^{2+}$ (control: $n = 5$, $N = 2$; Ca$^{2+}$-free: $n = 5$, $N = 2$; paired t test, $p = 0.005$). **b** Maximum intensity projection showing cortical neurons from an acute brain slice co-expressing GCaMP6f and mRGECO. **c** *Left*, Images depict GCaMP6f and mRGECO fluorescence from a patch-clamped pyramidal neuron during the application of a 50 Hz, 4-sec action potential train. The time (t) relative to stimulus onset is depicted. *Right*, Kymograph showing the change in GCaMP6f and mRGECO fluorescence along the somatodendritic axis. The ROIs used for producing the kymograph are labelled (**a–c**). **d** Sample trace showing that cytosolic Ca$^{2+}$ (GCaMP6f) in the soma and apical dendrite has a rapid stimulus-evoked rise time in comparison to mitochondrial Ca$^{2+}$ (mRGECO). **e** Stimulus-evoked changes in cytosolic Ca$^{2+}$ have a significantly shorter time to half peak than mitochondrial Ca$^{2+}$ in the soma (GCAMP: $n = 8$, $N = 5$; mRGECO: $n = 8$, $N = 5$; paired t test, $p = 0.005$) but not the apical dendrite (GCAMP: $n = 5$, $N = 3$; mRGECO: $n = 5$, $N = 3$; paired t test, $p = 0.23$). **f** Summary data showing the ratio of evoked changes in mitochondrial Ca$^{2+}$ and cytosolic Ca$^{2+}$ (coupling ratio) at 2-sec post-stimulus onset. The coupling ratio is low and not significantly different between the soma and apical dendrite (soma: $n = 5$, $N = 3$; dendrite: $n = 5$, $N = 3$; paired t test, $p = 0.126$). **g** *Left*, Sample Ca$^{2+}$ traces measured from the soma depict the rapid (seconds) recovery of post-train cytosolic Ca$^{2+}$ and the long-lasting (minutes) recovery of mitochondrial Ca$^{2+}$ to pre-stimulus baseline. The dashed lines depict the fitted monoexponential decay functions. *Right*, The decay time constant ($\tau$) acquired during the Ca$^{2+}$ recovery period is significantly longer in the mitochondria relative to the cytosol (GCAMP: $n = 5$, $N = 3$; mRGECO: $n = 5$, $N = 3$; paired t test, $p = 0.037$). Summary data presented as the mean ± SEM. *$P < 0.05$, **$P < 0.01$. The number of cell replicates is shown in the graphs as well as the 'n' value in the text. The number of animal replicates is represented by the 'N' value.

potentials[51,59,60]. We next characterized the relationship between cytosolic and mitochondrial Ca$^{2+}$ during action potential firing by simultaneously imaging mRGECO and cytosolic GCAMP6f (Fig. 2b, c), both of which have similar affinities for Ca$^{2+}$ (GCAMP6f $K_d = 375$ nM; mRGECO $K_d = 480$ nM)[48,61]. Delivering an action potential train (50 Hz, 4 sec) to pyramidal neurons co-expressing the two sensors caused an initial rapid increase in cytosolic Ca$^{2+}$ in the somatodendritic compartment followed by a latent slower rise in mitochondrial Ca$^{2+}$ (Fig. 2c–f; Supplementary Video 1). Following action potential firing, cytosolic Ca$^{2+}$ levels returned to baseline within seconds, while mitochondrial Ca$^{2+}$ recovered substantially more slowly, over minutes,

to pre-stimulus baseline (Fig. 2g). Thus, transient increases in cytosolic Ca$^{2+}$ may be capable of producing prolonged changes in mitochondrial function in the somatodendritic compartment that long outlast the period of action potential firing.

**MCU-dependent mitochondrial Ca$^{2+}$ uptake is tuned to action potential firing frequency.** In vivo, pyramidal neurons in the cortex and hippocampus typically have low spike firing rates (<5 Hz) but can enter states of high-frequency action potential firing (10-50 Hz) during sensory processing or other cognitive tasks[39,62–64]. Therefore, we next assessed the relationship between

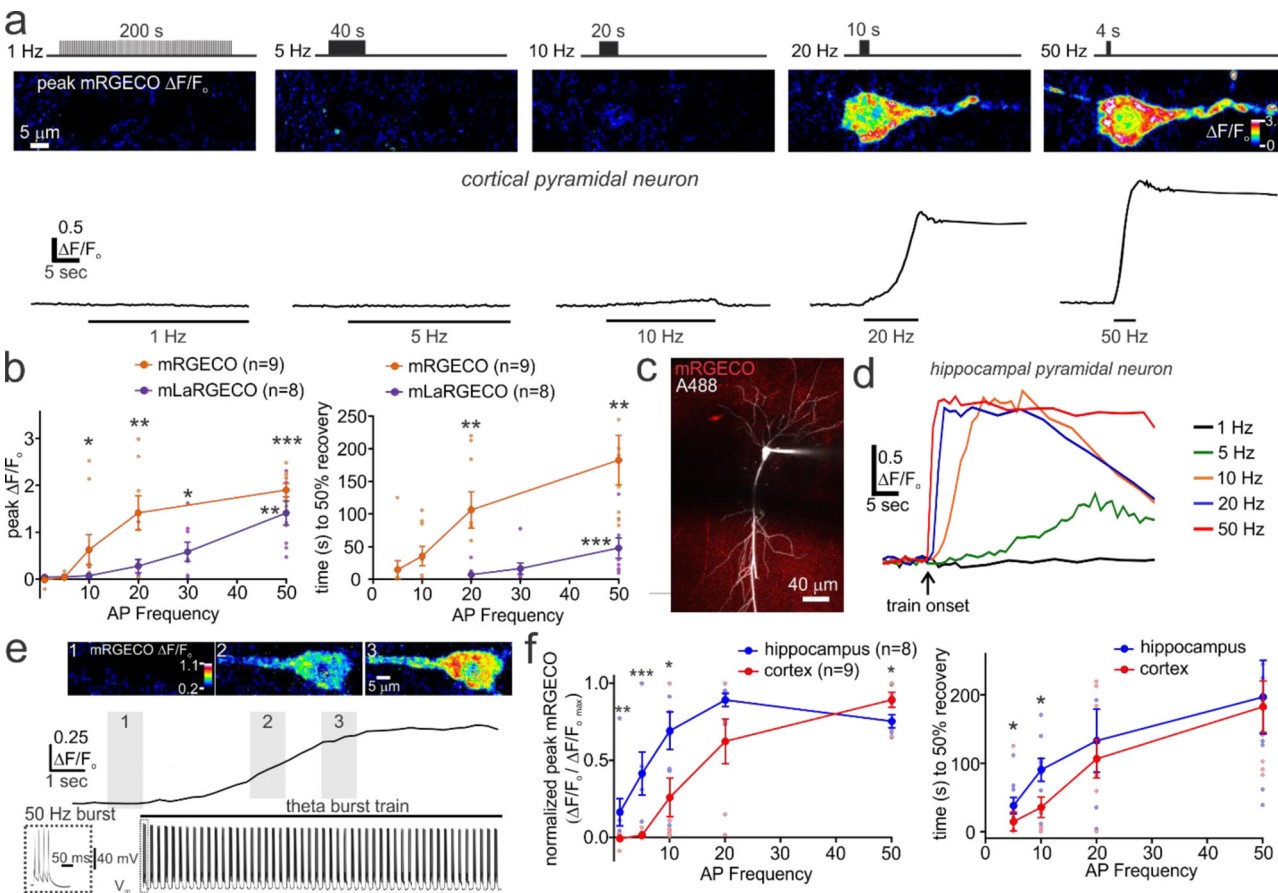

**Fig. 3 The coupling between activity and mitochondrial Ca²⁺ loading in pyramidal neurons is dependent on action potential firing frequency and varies between brain regions. a** *Upper*, Images depict the peak fluorescence change of mRGECO from a current-clamped cortical pyramidal neuron delivered action potential trains of progressively increasing frequencies. Stimuli of frequencies >5 Hz produced large and protracted elevations of mitochondrial Ca²⁺. All trains were comprised of 200 action potentials elicited by 200 current pulses (+2 nA, 5 ms). *Lower*, Traces depict changes in mRGECO fluorescence measured from the pyramidal neuron soma. **b** *Left*, Group data depicting the peak somatic fluorescence change of mRGECO and the low affinity sensor, mLaRGECO, in relation to action potential train frequency. Peak responses at each frequency were compared to the nominal responses elicited during 1-Hz trains (mRGECO data: $n = 9$, $N = 4$; Friedman's ANOVA and Dunn's multiple comparisons test: 5 Hz train, $p > 0.99$; 10 Hz train, $p = 0.029$; 20 Hz train, $p = 0.002$; 50 Hz train, $p = 0.0006$; mLaRGECO data: $n = 8$, $N = 3$; Friedman's ANOVA and Dunn's multiple comparisons test: 5 Hz train, $p > 0.99$; 10 Hz train, $p > 0.99$; 20 Hz train, $p = 0.224$; 30 Hz train, $p = 0.01$; 50 Hz train, $p = 0.003$). *Right*, Summary data showing the significantly greater time to recovery between low and high-frequency train-evoked fluorescence changes for mRGECO and mLaRGECO expressing neurons. For mRGECO experiments, statistical comparisons were made between responses at 5 Hz with all other frequencies ($n = 9$, $N = 4$; Friedman's ANOVA with Dunn's multiple comparison test: 10 Hz train, $p > 0.99$; 20 Hz train, $p = 0.0042$; 50 Hz train, $p = 0.0021$). For mLaRGECO experiments, statistical comparisons were made between the response to the 10 Hz train and all other frequencies ($n = 8$, $N = 3$; Friedman test with Dunn's multiple comparisons test; 20 Hz train, $p > 0.99$; 30 Hz train, $p = 0.0995$; 50 Hz train, $p = 0.0002$). **c** Patch-clamped and dye (A488) filled mRGECO expressing hippocampal CA1 pyramidal neuron. **d** Representative traces from a CA1 pyramidal neuron showing somatic mRGECO fluorescence changes in response to action potential firing frequencies greater than 1 Hz. **e** A CA1 pyramidal neuron showing progressive mitochondrial Ca²⁺ accumulation (*Upper*) when stimulated to fire action potential bursts applied at theta frequency (50 Hz bursts at 5 Hz) (*Lower*). **f** *Left*, Relationship between the train-evoked peak mRGECO responses and spike firing frequency in cortical (data set repeated from panel B) and hippocampal pyramidal neurons. The hippocampal data set included cells from the CA1 and CA3 subregions. For each neuron, the peak responses at each train frequency were normalized to the largest evoked response out of the series of action potential trains. Normalized peak responses are significantly greater in hippocampal vs cortical neurons during low frequency firing (cortex: $n = 9$, $N = 4$; hippocampus: $n = 8$, $N = 5$; Mann-Whitney test for all; 1 Hz train, $p = 0.001$; 5 Hz train, $p = 0.0003$; 10 Hz train, $p = 0.0345$; 20 Hz train, $p = 0.672$; 50 Hz train, $p = 0.015$). *Right*, The time to recovery from peak mRGECO response was significantly longer in neurons of the hippocampus relative to the cortex during the 5 and 10 Hz action potential trains (5 Hz train, cortex: $n = 9$, $N = 4$; hippocampus: $n = 9$, $N = 5$; Mann-Whitney test, $p = 0.0142$; 10 Hz train, cortex: $n = 9$, $N = 4$; hippocampus: $n = 9$, $N = 5$; Mann-Whitney test, $p = 0.0244$; 20 Hz train, cortex: $n = 9$, $N = 4$; hippocampus: $n = 7$, $N = 5$; unpaired t test, $p = 0.611$; 50 Hz train, cortex: $n = 8$, $N = 4$; hippocampus: $n = 8$, $N = 5$; unpaired t test, $p = 0.71$). Summary data presented as the mean ± SEM. * $P < 0.05$, **$P < 0.01$, *** $P < 0.001$. The number of cell replicates is shown in the graphs as well as the 'n' value in the text. The number of animal replicates is represented by the 'N' value.

spike frequency and mitochondrial Ca²⁺ uptake to determine the relevance of the MCU to discrete firing intensities. mRGECO expressing cortical pyramidal neurons were patch-clamped and action potential trains were evoked at progressively increasing frequencies (1, 5, 10, 20, and 50 Hz). During low-frequency action

potential trains (1 & 5 Hz), no apparent changes in the somatic mitochondrial Ca²⁺ signals were detected throughout the duration of stimulus (Fig. 3a, b). In contrast, the same number of action potentials applied at 10 Hz or greater resulted in rapid and large increases in mitochondrial Ca²⁺ (Fig. 3a, b). These

responses became progressively larger and more prolonged with higher action potential firing frequency (Fig. 3a, b). The relationship between the peak mitochondrial $Ca^{2+}$ response and spike frequency also appeared to plateau during high-frequency activity. This suggests that mitochondrial $Ca^{2+}$ levels become constrained or that mREGCO saturates at the greater $Ca^{2+}$ concentrations reached during high-frequency stimulation. To test the latter possibility, we repeated these experiments using the low-affinity mitochondrial $Ca^{2+}$ sensor, mitoLaRGECO1.2 (mLaRGECO) ($K_d = 12\,\mu M$)[48]. Action potential evoked mLaRGECO fluorescence changes were smaller and briefer in duration relative to those measured with the high-affinity mRGECO sensor (Fig. 3b). Moreover, the spike frequency-response relationship measured with mLaRGECO was non-sigmoidal and lacked a clear plateau at high-frequency stimulation. This indicates that $Ca^{2+}$ saturation of mRGECO likely contributed to the relationship between spike frequency and response magnitude (Fig. 3b). Collectively, these results demonstrate that MCU activation and subsequent mitochondrial $Ca^{2+}$ uptake are highly sensitive to accelerated spike firing frequency in cortical pyramidal neurons.

**The coupling between neuronal activity and mitochondrial $Ca^{2+}$ uptake varies between brain regions.** Anatomically and functionally discrete brain regions can exhibit distinct neuroenergetic coupling, $Ca^{2+}$ channel subtypes, and suite of mitochondrial proteins[41–43]. Therefore, it may be possible for the relationship between neuronal activity and mitochondrial $Ca^{2+}$ uptake to differ between distinct cell populations. To address this, we patch clamped mRGECO expressing hippocampal pyramidal neurons and compared the evoked responses to those measured in cortical neurons (Fig. 3c). Similar to cortical neurons, action potential trains in these neurons produced progressively greater mitochondrial $Ca^{2+}$ uptake with higher action potential firing frequencies (Fig. 3d). We also assessed whether mitochondrial $Ca^{2+}$ uptake occurred in response to activity patterns that are relevant to hippocampal neuron activity in vivo. To do this, we imaged CA1 pyramidal neurons while triggering theta burst firing. This stimulus pattern mimics the high-frequency action potential bursts (>50 Hz) that occur at theta frequency (~5 Hz) in CA1 neurons during rodent exploratory behaviour and learning processes[62,63,65,66]. When CA1 neurons were triggered to fire high-frequency bursts applied at theta frequency, somatodendritic mitochondrial $Ca^{2+}$ increased within seconds after the stimulus onset and remained elevated above pre-stimulus baseline for an extended period of time after the train (peak $\Delta F/F_o$: $1.19 \pm 0.26$; time (sec) to 50% recovery: $87.41 \pm 20.64$, $n = 4$, $N = 3$) (Fig. 3e). This evidence supports a role for mitochondrial $Ca^{2+}$ uptake during behavioural tasks associated with high-frequency signalling and theta bursting in vivo. The relationship between activity and mitochondrial $Ca^{2+}$ uptake revealed that the peak train-evoked mRGECO fluorescence responses in hippocampal neurons plateaued at high action potential frequencies, similar to cortical neurons (Fig. 3f). Interestingly, low-frequency spike firing was much more effective at eliciting mitochondrial $Ca^{2+}$ uptake in hippocampal neurons relative to those in the cortex (Fig. 3f). Consistent with this, the time to recovery from peak mitochondrial $Ca^{2+}$ was also longer at low firing frequency in pyramidal neurons of the hippocampus relative to the cortex (Fig. 3f). These differences demonstrate a brain region-dependent variation in the strength of coupling between action potential firing and mitochondrial $Ca^{2+}$ uptake by the MCU.

**Adaptation of mitochondrial energy metabolism in pyramidal neurons during high-frequency action potential firing is mediated by the MCU.** Mitochondrial $Ca^{2+}$ uptake can depolarize the mitochondrial membrane potential, facilitate the activity of the electron transport chain, and promote the $Ca^{2+}$-dependent activation of several key enzymes of the tricarboxylic acid cycle[30,67–69]. We therefore examined how MCU-dependent mitochondrial $Ca^{2+}$ flux controlled action potential evoked metabolic responses in single pyramidal neurons using two-photon microscopy of intracellular nicotinamide adenine dinucleotide hydride (NADH) autofluorescence[70–72]. NADH is a coenzyme that is generated from the reduction of non-fluorescent $NAD^+$ by the mitochondrial TCA cycle or via glycolysis in the cytosol[73]. NADH is oxidized to $NAD^+$ at complex I of the electron transport chain in order to generate the proton gradient used to synthesize ATP[73] (Fig. 4a). NADH autofluorescence can thus be used to assess changes in neuronal metabolic function. The measured autofluorescence will be referred to as NAD(P)H to acknowledge the inability to discern NADH from NADPH autofluorescence, although the former is more abundant in brain tissue[73,74].

Two-photon excitation of acute cortical brain slices revealed NAD(P)H autofluorescence in neuronal cell bodies that was distributed in a punctate pattern, similar to mitochondria labelled with mRGECO (Fig. 4b). We next performed patch-clamp recordings of cortical pyramidal neurons and measured NAD(P)H autofluorescence during the range of firing frequencies used to study mitochondrial $Ca^{2+}$ uptake in our prior experiments. At firing frequencies above 1 Hz, there was a notable dip in NAD(P)H autofluorescence which became more prominent at higher action potential firing frequency (Fig. 4c–e). This dip reflects the oxidation of mitochondrial NADH by the electron transport chain in response to neuronal activation[30,69,73]. At low firing frequencies, the NAD(P)H autofluorescence returned to pre-stimulus baseline shortly after the end of the stimulus (Fig. 4c, e). Conversely, after high-frequency action potential firing (20 & 50 Hz), the transient dip in NAD(P)H autofluorescence was followed by a long-lasting overshoot that eventually recovered to baseline after a minute or more (Fig. 4c). This NAD(P)H overshoot is consistent with a prolonged chemical reduction of $NAD^+$ to NADH, which can result from mitochondrial TCA cycle activity or glycolysis in the cytosol[75,76]. As the low-frequency trains were longer than the high-frequency stimuli, it may be possible that evoked metabolic responses are more likely to be influenced by prolonged laser exposure. We therefore examined whether NAD(P)H transients triggered by a 50-Hz train were influenced by 5 min of pre-exposure to the excitation laser at the illumination power used for imaging. Conditioning cells with the laser had no effect on the train-evoked NAD(P)H decrease or the long-lasting overshoot phase, suggesting that laser illumination had no influence on our results (Fig. 4f).

To test the involvement of mitochondrial metabolism, we examined the sensitivity of evoked NAD(P)H transients to rotenone ($10\,\mu M$), which prevents NADH oxidation by complex I of the electron transport chain[77]. Wash in of rotenone caused an elevation of neuronal autofluorescence in unstimulated neurons, consistent with a disruption of NADH oxidation in the presence of ongoing TCA cycle activity (Fig. 5a)[78]. Subsequent stimulation of neurons in the presence of rotenone largely prevented the stimulus-evoked NAD(P)H transients (Fig. 5a, b). This data supports the interpretation that the measured autofluorescence changes primarily reflect mitochondrial NAD(P)H dynamics.

The growth in the magnitude of evoked NAD(P)H transients with increasing firing frequency paralleled the previously measured occurrence of mitochondrial $Ca^{2+}$ uptake. We next examined a role for mitochondrial $Ca^{2+}$ uptake in generating these NAD(P)H signals by blocking the MCU with Ru360 dialysis. Under these conditions, applying a 50-Hz, 4-sec train produced a briefer train-evoked dip in NAD(P)H

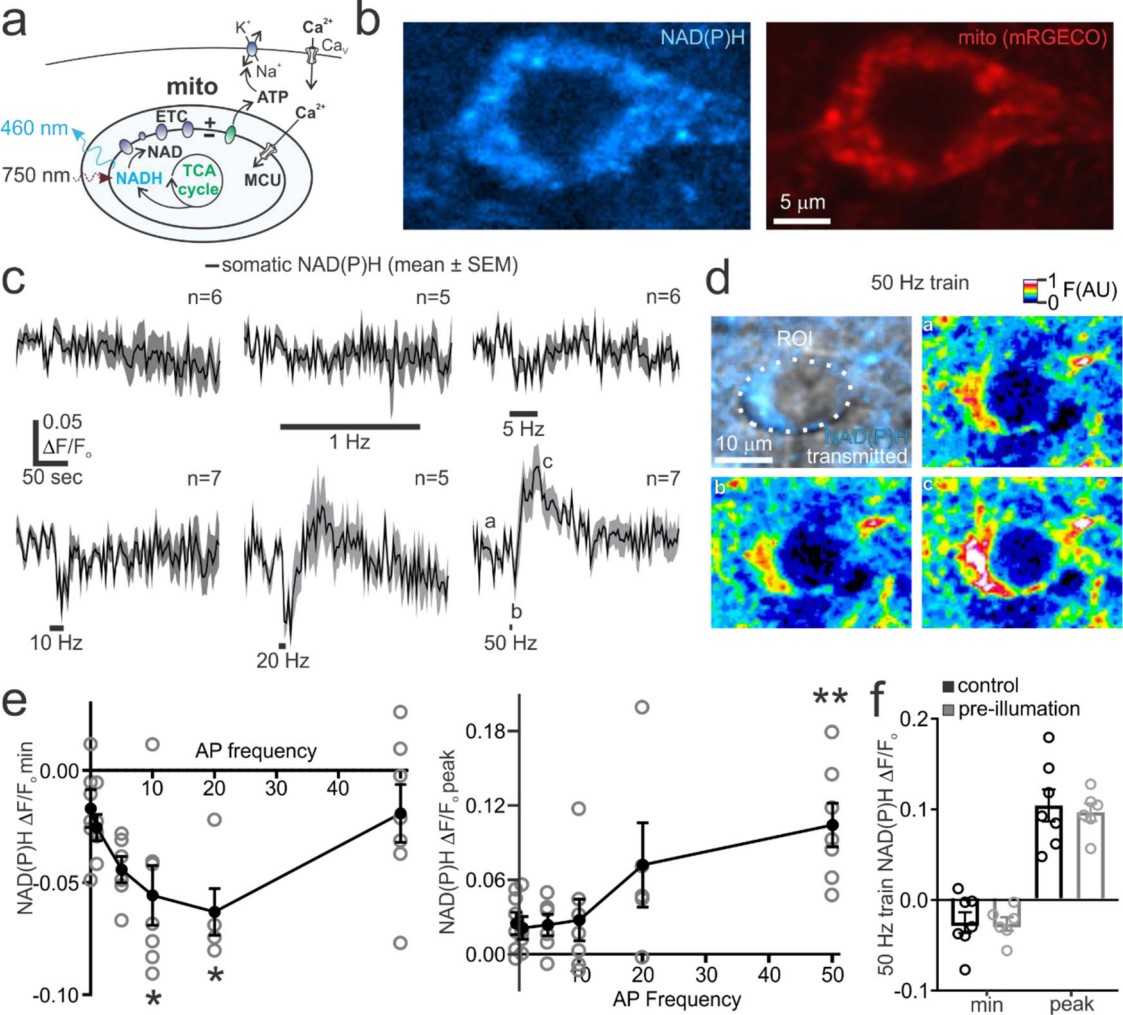

**Fig. 4 Measurement of action potential evoked NAD(P)H dynamics in single pyramidal neurons in situ. a** *Left*, Illustration depicts the generation of fluorescent NADH by the TCA cycle and its oxidation to non-fluorescent NAD$^+$ by the electron transport chain (ETC). NADH autofluorescence can be measured using two-photon excitation at 750 nm. **b** Images of a cortical pyramidal neuron showing the punctate pattern of intracellular NAD(P)H autofluorescence and mitochondria labelled with mRGECO. **c** Group time series data showing the enlargement of somatic NAD(P)H transients with progressively higher action potential firing frequencies in a cortical pyramidal neuron. Labels during the 50 Hz train highlight the pre-stimulus phase (**a**), the stimulus-evoked transient decrease (**b**), and the long-lasting elevation of NAD(P)H autofluorescence (**c**). Images from these time points are depicted in panel D. **d** NAD(P)H autofluorescence in a pyramidal neuron soma during the baseline (**a**), immediately following a 50 Hz, 4-sec action potential train (**b**), and tens of seconds post-stimulus (**c**). The region of interest (ROI) used for analysis is depicted. **e** Relative to unstimulated neurons ($n = 6$, $N = 4$) train-evoked evoked NAD(P)H dips (min) (*Left*) and transient peaks (*Right*) significantly increase in magnitude with higher action potential firing frequency (NAD(P)H min data, One-way ANOVA and Dunnett's multiple comparisons test: 1 Hz train, $n = 5$, $N = 2$, $p = 0.976$; 5 Hz train, $n = 6$, $N = 4$, $p = 0.274$; 10 Hz, $n = 7$, $N = 3$, $p = 0.05$; 20 Hz, $n = 5$, $N = 2$, $p = 0.028$; 50 Hz, $n = 7$, N $= 4$, $p = 0.99$; NAD(P)H peak stats, One-way ANOVA and Dunnett's multiple comparisons test: 1 Hz train, $n = 5$, $N = 2$, $p = 0.99$; 5 Hz train, $n = 6$, $N = 4$, $p > 0.999$; 10 Hz, $n = 7$, $N = 3$, $p = 0.99$; 20 Hz, $n = 5$, $N = 2$, p $= 0.263$; 50 Hz, $n = 7$, $N = 4$, $p = 0.0097$. **f** Prior exposure of neurons to 5 min of 750 nm excitation light had no significant effect on NAD(P)H dips or latent peaks triggered by a 50 Hz train of action potentials (NAD(P)H min data, control: $n = 7$, $N = 4$; pre-illumination: n $= 6$, N $= 4$; unpaired t test, $p = 0.92$; NAD(P)H peak data, control: $n = 7$, $N = 4$; preillumination: $n = 6$, $N = 4$; unpaired $t$ test, $p = 0.72$). Summary data presented as the mean ± SEM. *$P < 0.05$, **$P < 0.01$. The number of cell replicates is shown in the graphs as well as the 'n' value in the text. The number of animal replicates is represented by the 'N' value.

autofluorescence relative to controls (Fig. 5a, b). Moreover, the latent NAD(P)H overshoot phase was substantially depressed in the presence of Ru360. These data indicate that the MCU has a prominent role in modulating NADH oxidation and the reduction of NAD$^+$ to NADH during high-frequency action potential firing.

**Mitochondrial Ca$^{2+}$ uptake buffers cytosolic Ca$^{2+}$ and reduces the slow afterhyperpolarization duration during high-frequency action potential firing.** The detected rise in mitochondrial Ca$^{2+}$ during high-frequency activity implies that mitochondrial Ca$^{2+}$

uptake may shape the magnitude and temporal dynamics of cytosolic Ca$^{2+}$ in the somatodendritic compartment[9]. We therefore assessed the extent to which MCU-dependent Ca$^{2+}$ uptake influenced the relationship between cortical pyramidal neuron action potential firing frequency and evoked cytosolic Ca$^{2+}$. This was accomplished using simultaneous whole-cell current-clamp recordings and measurement of cytosolic Ca$^{2+}$ with the low-affinity Ca$^{2+}$-sensitive dye Fluo5N. In control conditions there was an increase in the Ca$^{2+}$ rise magnitude with progressively greater action potential firing frequencies (Fig. 6a). The influence of mitochondrial Ca$^{2+}$ buffering was tested by dialyzing neurons

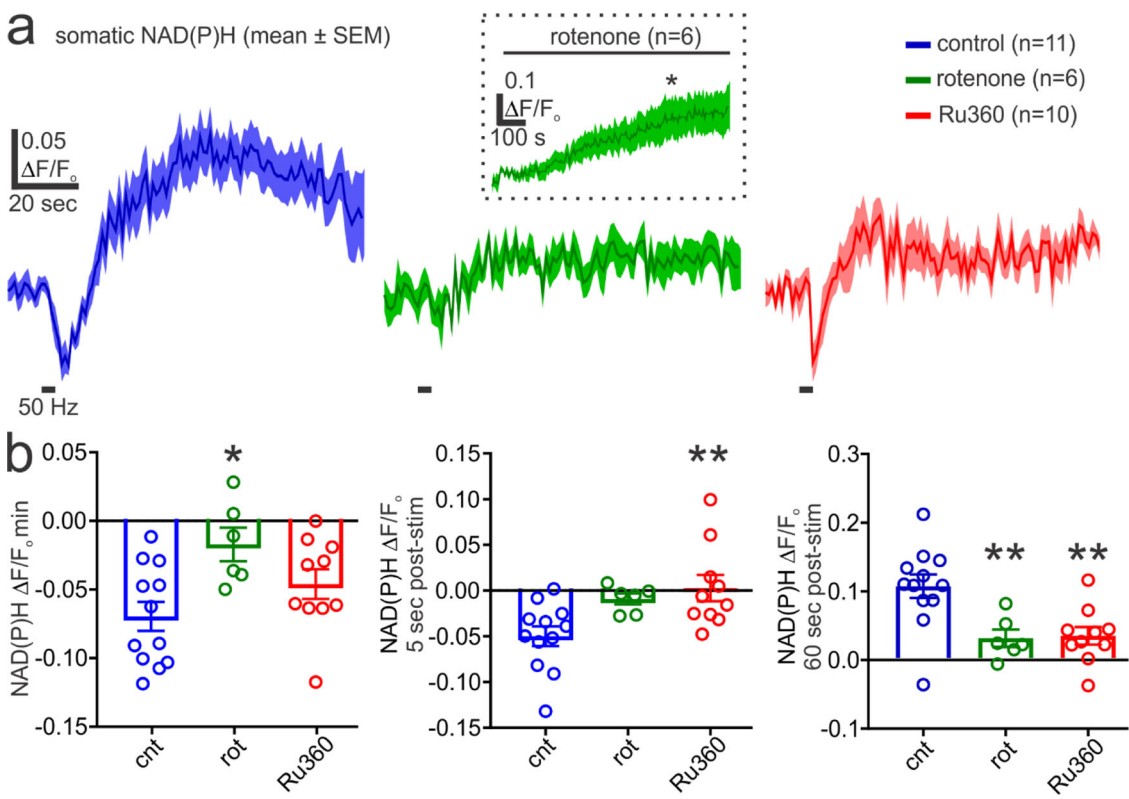

**Fig. 5 The MCU governs NAD(P)H dynamics during high-frequency action potential firing in pyramidal neurons. a** *Left & Middle*, Relative to control, blocking mitochondrial complex I with rotenone (10 μM) occludes the NAD(P)H dip and delayed overshoot evoked by a 50 Hz action potential train. *Inset*, Bath application of rotenone produces a significant increase in NAD(P)H fluorescence relative to baseline in an unstimulated neuron ($n = 6$, $N = 5$; paired t test, $p = 0.036$). *Right*, Dialysis of Ru360 (20 μM), to block MCU-dependent mitochondrial $Ca^{2+}$ uptake, partially attenuates the stimulus-evoked NAD(P) H dip and largely prevents the delayed overshoot phase. Traces from each condition represent the mean and SEM of NAD(P)H measured from multiple pyramidal neurons. **b** *Left*, Rotenone, but not Ru360, significantly reduces the peak train-evoked NAD(P)H dip (control: $n = 12$, $N = 6$; rotenone: $n = 6$, $N = 3$; Ru360: $n = 10$, $N = 5$; One-way ANOVA and Dunnett's multiple comparisons test: rotenone, $p = 0.01$; Ru360, $p = 0.22$). *Middle*, The NAD(P)H dip magnitude at 5 sec post-stimulus is significantly attenuated in Ru360 relative to control (control: $n = 12$, $N = 6$; rotenone: $n = 6$, $N = 3$; Ru360: $n = 10$, $N = 5$; One-way ANOVA and Dunnett's multiple comparisons test: rotenone, $p = 0.073$; Ru360, $p = 0.0059$). *Right*, The evoked NAD(P)H overshoot, measured at 60-sec post-train, is significantly reduced by rotenone or Ru360 (control: $n = 12$, $N = 6$; rotenone: $n = 6$, $N = 3$; Ru360: $n = 10$, $N = 5$; One-way ANOVA and Dunnett's multiple comparisons test: rotenone, $p = 0.0086$; Ru360, $p = 0.0037$). Summary data presented as the mean ± SEM. *$P < 0.05$, **$P < 0.01$. The number of cell replicates is shown in the graphs as well as the 'n' value in the text. The number of animal replicates is represented by the '$N$' value.

with Ru360 (20 μM) to inhibit the MCU. At a low action potential firing frequency of 5 Hz there was no apparent influence of Ru360 on evoked cytosolic $Ca^{2+}$ dynamics relative to control conditions (Fig. 6a, c). However, at action potential firing frequencies of 10 Hz or greater, Ru360 increased the peak cytosolic $Ca^{2+}$ magnitude and time for post-train $Ca^{2+}$ recovery (Fig. 6a–c; Supplementary Video 2). The most parsimonious explanation for these effects of Ru360 on cytosolic $Ca^{2+}$ signals is disruption of the MCU, as this drug does not affect other determinants of $Ca^{2+}$ signalling, such as the $Na^+/Ca^{2+}$ exchanger or voltage-gated $Ca^{2+}$ channels[56]. Consistent with this, the effects of Ru360 on cytosolic $Ca^{2+}$ selectively occurred at firing frequencies which were demonstrated to evoke mitochondrial $Ca^{2+}$ uptake with mRGECO. Measurement of the membrane voltage revealed the progressive growth in the magnitude of the post-train slow afterhyperpolarization (sAHP) with greater cytosolic $Ca^{2+}$ and higher action potential firing frequency (Fig. 6a, c). The sAHP is mediated by the $Ca^{2+}$-dependent activation of a neuromodulator-sensitive voltage-independent $K^+$ current[46]. This current is present in cells of multiple brain regions, where it plays a critical role in shaping neuronal signalling by promoting negative feedback control of excitability[46]. Interestingly, blocking mitochondrial $Ca^{2+}$ uptake with Ru360 led to a prolonged sAHP relative to

controls, selectively during high frequency action potential firing (Fig. 6a, c). The prolongation of the sAHP closely correlated with the enlarged magnitude and protracted temporal dynamics of cytosolic $Ca^{2+}$ elevations in the presence of Ru360. The dramatic effect of Ru360 on cytosolic $Ca^{2+}$ and the sAHP kinetics suggests that mitochondrial $Ca^{2+}$ buffering strongly controls the activation of the sAHP current ($I_{sAHP}$), and therefore neuronal excitability.

**Mitochondrial $Ca^{2+}$ buffering reduces the coupling between $Ca^{2+}$ influx and activation of the sAHP current in pyramidal neurons.** To examine the interplay between mitochondrial $Ca^{2+}$ uptake and the sAHP, we next measured $I_{sAHP}$ in hippocampal CA1 pyramidal neurons, as it is well characterized in these cells[46]. Under whole-cell voltage-clamp, applying a series of progressively longer transient step depolarizations, to initiate voltage-gated $Ca^{2+}$ influx, resulted in the serial enlargement of the $I_{sAHP}$ (Fig. 7a, b). The $I_{sAHP}$ had a linear relationship with membrane voltage and a reversal potential consistent with a $K^+$ conductance (Fig. 7c, d). In pyramidal neurons in which the MCU was blocked by dialysis with Ru360, there was no appreciable difference in the $I_{sAHP}$ magnitude evoked by brief step

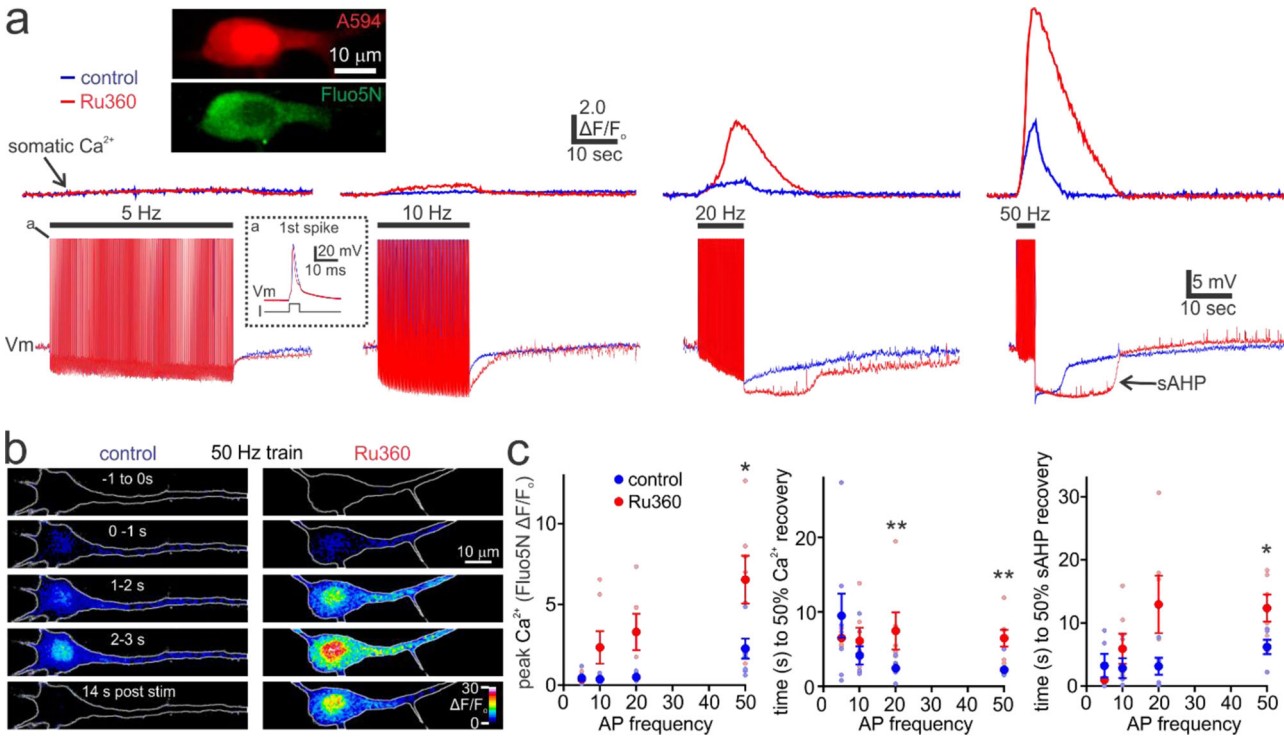

**Fig. 6 Mitochondrial Ca$^{2+}$ uptake clamps cytosolic Ca$^{2+}$ magnitude and reduces the slow afterhyperpolarization duration elicited by high-frequency action potential firing. a** Representative traces of cytosolic Ca$^{2+}$ in the soma measured with Fluo5N (*Upper*), and membrane potential (Vm) (*Lower*), from patch-clamped pyramidal neurons during low and high-frequency action potential trains. Upper inset depicts a pyramidal neuron filled with Fluo5N and the morphology marker Alexa594 (A594). Lower inset shows a single action potential from neurons in control and Ru360 conditions. Ru360 dialysis results in substantially larger and more prolonged cytosolic Ca$^{2+}$ responses selectively during high-frequency action potential trains. The train-evoked slow afterhyperpolarization (sAHP) increases in duration with high-frequency trains and is significantly prolonged relative to control when the MCU is blocked by Ru360. **b** Images depict the change in cytosolic Ca$^{2+}$ (Fluo5N ΔF/F$_o$) in pyramidal neurons before, during, and after a 50 Hz, 4-sec train of action potentials in the presence (control) and absence of mitochondrial Ca$^{2+}$ buffering (Ru360). The somatodendritic Ca$^{2+}$ responses were substantially larger and more prolonged in the absence of mitochondrial Ca$^{2+}$ buffering. **c** *Left*, Ru360 significantly increased the peak cytosolic Ca$^{2+}$ response evoked during high but not low frequency action potential firing (5 Hz train, control: $n = 9$, $N = 5$; Ru360: $n = 7$, $N = 4$; unpaired t test with Welch's correction, $p = 0.24$; 10 Hz train, control: $n = 8$, $N = 4$; Ru360: $n = 7$, $N = 4$; Mann-Whitney test, $p = 0.12$; 20 Hz train, control: $n = 8$, $N = 5$; Ru360: $n = 6$, $N = 4$; unpaired t test with Welch's correction, $p = 0.057$; 50 Hz train, control: $n = 8$, $N = 4$; Ru360: $n = 7$, $N = 4$; Mann-Whitney test, $p = 0.04$). *Middle*, Ru360 treatment results in a significantly greater time to 50% recovery from peak Ca$^{2+}$ response at high frequency firing rates (5 Hz train, control: $n = 8$, $N = 5$; Ru360: $n = 5$, $N = 4$; unpaired t test with Welch's correction, $p = 0.35$; 10 Hz train, control: $n = 7$, $N = 4$; Ru360: $n = 7$, $N = 4$; Mann-Whitney test, $p = 0.23$; 20 Hz train, control: $n = 8$, $N = 5$; Ru360: $n = 6$, $N = 4$; Mann-Whitney test, $p = 0.0047$; 50 Hz train, control: $n = 8$, $N = 4$; Ru360: $n = 7$, $N = 4$; Mann-Whitney test, $p = 0.002$). *Right*, Compared to controls, Ru360 significantly increased the time to 50% recovery from the sAHP following high frequency action potential firing (5 Hz train, control: $n = 6$, $N = 4$; Ru360: $n = 6$, $N = 4$; Mann-Whitney test, $p = 0.94$; 10 Hz train, control: $n = 6$, $N = 6$; Ru360: $n = 6$, $N = 4$; Mann-Whitney test, $p = 0.23$; 20 Hz train, control: $n = 6$, $N = 5$; Ru360: $n = 6$, $N = 4$; unpaired t test with Welch's correction, $p = 0.12$; 50 Hz train, control: $n = 8$, $N = 6$; Ru360: $n = 6$, $N = 4$; unpaired t test, $p = 0.017$). Summary data presented as the mean ± SEM. *$P < 0.05$, **$P < 0.01$. The number of cell replicates is shown in the graphs as well as the 'n' value in the text. The number of animal replicates is represented by the 'N' value.

depolarizations (Fig. 7a, b). However, with progressively longer step depolarizations (>600 ms), the I$_{sAHP}$ in Ru360 became larger than control conditions (Fig. 7a, b). Examining the I$_{sAHP}$ current-voltage relationship revealed that the depolarization evoked K$^+$ current was sustained for a longer time in neurons exposed to Ru360, relative to controls (Fig. 7c, d). These findings indicate that Ru360 substantially enhances the activation of the Ca$^{2+}$-activated K$^+$ current underlying the I$_{sAHP}$, likely because of the reduced Ca$^{2+}$ buffering of cytosolic Ca$^{2+}$ when the MCU is blocked. We tested this possibility by examining the effect of Ru360 on the I$_{sAHP}$ when pyramidal neuron Ca$^{2+}$ buffering was enhanced by intracellular dialysis of the Ca$^{2+}$ chelator BAPTA (10 mM). Under these conditions, BAPTA effectively eliminated the depolarization-evoked I$_{sAHP}$ and largely prevented the enhancement of the current by Ru360 (Fig. 7e, f). These results are consistent with mitochondrial Ca$^{2+}$ buffering preventing the Ca$^{2+}$-dependent activation of the I$_{sAHP}$ in pyramidal neurons.

## Discussion

Mitochondrial Ca$^{2+}$ uptake mediated by the MCU has long been implicated as a regulator of energy metabolism and intracellular Ca$^{2+}$ signalling in neurons. Despite this, the functional relevance of MCU activation to neuronal activity in the brain has remained unclear. Using in situ two-photon imaging and electrophysiology, we revealed that: (1) The degree of mitochondrial Ca$^{2+}$ uptake is a function of spike firing frequency in pyramidal neurons; (2) Increasing spike frequencies coupled with greater MCU-dependent Ca$^{2+}$ uptake led to adaptive mitochondrial energy metabolism; (3) The coupling between action potential firing and mitochondrial Ca$^{2+}$ uptake varied between brain regions; (4) Ca$^{2+}$ uptake by mitochondria provides a remarkable degree of cytoplasmic Ca$^{2+}$ buffering in pyramidal neurons during high-frequency firing and thereby controlled sAHP magnitude. We propose that MCU activation facilitates pyramidal neuron signalling by enhancing energy metabolism and excitability during

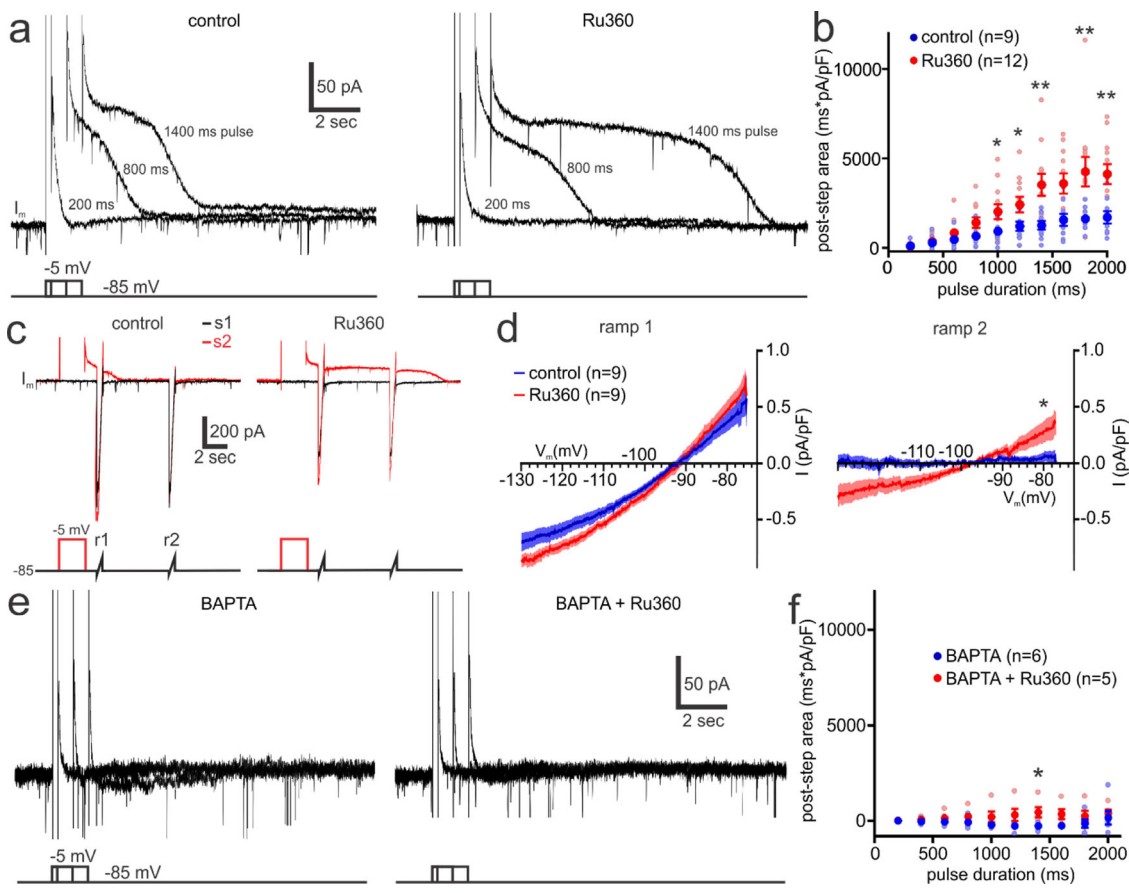

**Fig. 7 Mitochondrial Ca²⁺ buffering reduces the coupling between Ca²⁺ influx and activation of the sAHP current in pyramidal neurons. a** Traces depict the sAHP membrane current (I_sAHP) in a patch-clamped hippocampal CA1 pyramidal neuron in response to step depolarizations (−85 mV to −5 mV) of progressively increasing duration. Blocking mitochondrial Ca²⁺ uptake with Ru360 substantially enlarged the I_sAHP evoked by long, but not short, step depolarizations. **b** Summary data showing that Ru360 significantly increased the evoked outward current area following long depolarizing steps (control: $n = 9$, $N = 9$; Ru360: $n = 12$, $N = 9$; 200 ms pulse, unpaired t test, $p = 0.849$; 400 ms pulse, unpaired t test, $p = 0.705$; 600 ms pulse, unpaired t test, $p = 0.207$; 800 ms pulse, unpaired t test, p = 0.064; 1000 ms pulse, unpaired t test with Welch's correction, $p = 0.0365$; 1200 ms pulse, unpaired t test, $p = 0.041$; 1400 ms pulse, unpaired t test with Welch's correction, $p = 0.004$; 1600 ms pulse, unpaired t test, $p = 0.01$; 1800 ms pulse, unpaired t test with Welch's correction, $p = 0.009$; 2000 ms pulse, unpaired t test, $p = 0.003$). **c** Membrane currents in control and Ru360 produced by two voltage ramps (r1 & r2) from -135 to −75 mV, measured with (sweep 2 [s2]) and without (sweep 1 [s1]) a preceding step depolarization. **d** Current-voltage (I-V) relationship for the I_sAHP, measured from ramp 1 and ramp 2, in the presence and absence of Ru360. The I_sAHP current has a reversal potential consistent with a potassium current and is significantly larger in Ru360 (ramp 2 current at −80 mV, control: $n = 9$, $N = 9$; Ru360: $n = 9$, $N = 8$; unpaired t test, $p = 0.049$). The I-V plot for the I_sAHP was produced by subtracting the voltage-ramp currents of sweep 1 from sweep 2. **e** Increasing the Ca²⁺ buffering capacity of the cytosol via intracellular dialysis of 10 mM BAPTA largely prevented the enhancement of the evoked I_sAHP produced by disrupting mitochondrial Ca²⁺ buffering with Ru360. **f** In the presence of intracellular BAPTA, Ru360 had no significant effect on the evoked outward current area, except during the 1400 ms step depolarization (BAPTA: $n = 6$, $N = 3$; BAPTA + Ru360: $n = 5$, $N = 2$; 200 ms pulse, unpaired t test, $p = 0.159$; 400 ms pulse, unpaired t test, $p = 0.22$; 600 ms pulse, unpaired t test, $p = 0.23$; 800 ms pulse, unpaired t test, $p = 0.18$; 1000 ms pulse, unpaired t test with Welch's correction, $p = 0.22$; 1200 ms pulse, Mann-Whitney test, $p = 0.082$; 1400 ms pulse, Mann-Whitney test, $p = 0.03$; 1600 ms pulse, unpaired *t* test, $p = 0.065$; 1800 ms pulse, unpaired t test, $p = 0.29$; 2000 ms pulse, Mann-Whitney test, $p = 0.247$). Summary data presented as the mean ± SEM. *$P < 0.05$, **$P < 0.01$, ***$P < 0.001$. The number of cell replicates is shown in the graphs as well as the 'n' value in the text. The number of animal replicates is represented by the 'N' value.

the brain processes associated with high-frequency action potential firing (Fig. 8).

Using a genetically encoded mitochondrial Ca²⁺ sensor, our experiments revealed the relationship between pyramidal neuron firing frequency and mitochondrial Ca²⁺ uptake in situ. Elevations in mitochondrial Ca²⁺ were largely absent during low-frequency action potential firing, but rapidly became larger and more prolonged in response to high-frequency trains or physiologically relevant bursting patterns. This relationship was also described in another recent study on cortical pyramidal neurons in brain slices[79]. As the in vitro K_d of mRGECO is similar (0.48 μM) to the resting mitochondrial Ca²⁺ concentration[48,80],

the lack of responsiveness at low activity levels likely reflects a true absence of Ca²⁺ uptake rather than unresponsiveness of the sensor. The apparent threshold level of action potential firing frequency required for mitochondrial Ca²⁺ uptake in cortical pyramidal neurons is consistent with the established relationship between cytosolic Ca²⁺ and MCU activation. This relationship shows a threshold and positive cooperativity, properties which are controlled by the Ca²⁺-sensitive MICU1 and MICU2 proteins[11,43]. The Ca²⁺ threshold of MCU activation likely also contributed to the observed temporal dissociation between the rise in cytosolic and mitochondrial Ca²⁺ levels during the onset of action potential firing. While the specific Ca²⁺ concentration

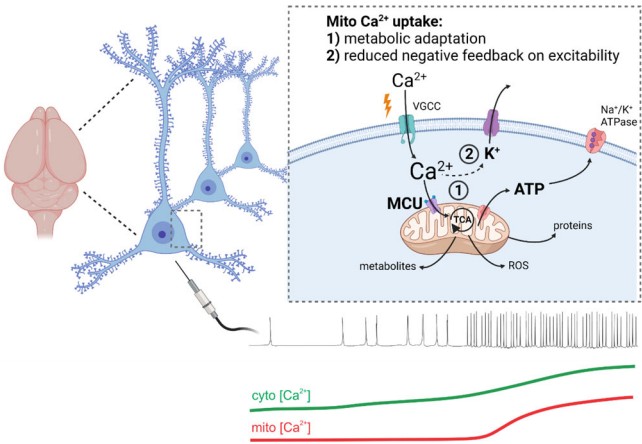

**Fig. 8 A functional model for mitochondrial Ca²⁺ uptake during shifts to high-frequency action potential firing in pyramidal neurons.** As pyramidal neurons enter states of high-frequency action potential firing, cytosolic $Ca^{2+}$ first rises as a result of voltage-gated $Ca^{2+}$ influx and subsequently achieves the threshold $Ca^{2+}$ concentration required to trigger MCU activation. (1) Mitochondrial $Ca^{2+}$ uptake facilitates electron transport chain activity and induces a protracted activation of the TCA cycle leading to increased production of ATP. (2) Simultaneously, the removal of cytosolic $Ca^{2+}$ by the mitochondria reduces the coupling between $Ca^{2+}$ influx and the activation of the $I_{AHP}$, which mediates negative feedback control of neuronal excitability. Thus, MCU-dependent mitochondrial $Ca^{2+}$ uptake adapts energy metabolism and excitability in a manner that is expected to sustain activity during periods of accelerated action potential firing rate underlying information processing by the brain. *Illustration was adapted from templates provided by BioRender.*

required for MCU activation in pyramidal neurons was not determined in our study, it has been estimated to be ~1 µM in non-neuronal cells[11]. However, this threshold is likely to be lower in neurons, due to the abundance of MICU3, which enhances MCU activation[2,43]. Additionally, recent in vivo imaging of pyramidal neurons provided evidence that mitochondrial $Ca^{2+}$ uptake evoked by brain activity is facilitated by CaMKII function, which is sensitive to $Ca^{2+}$ signal frequency and duration[37,81]. Thus, such factors may govern the relationship between spike firing frequency and MCU activation, in addition to the absolute magnitude of cytosolic $Ca^{2+}$ elevations.

Using single-cell imaging of intracellular NAD(P)H and electrophysiology we found that overt changes in adaptive energy metabolism increased in relation to spike firing frequency and paralleled the uptake of mitochondrial $Ca^{2+}$. Moreover, mitochondrial $Ca^{2+}$ uptake by the MCU governed NAD(P)H transients during high-frequency signalling. The partial influence of the MCU on the activity evoked NAD(P)H dip is consistent with enhanced NADH oxidation by the electron transport chain as a result of $Ca^{2+}$ entry dissipating the mitochondrial membrane potential[30,69,78]. As NADH oxidation was not eliminated by MCU blockade, electron transport chain activity is likely also facilitated by ADP produced by activity-dependent ATP consumption[69]. The MCU dependence of the latent NAD(P)H overshoot phase indicates that this channel is also critical for increasing the mitochondrial $NADH/NAD^+$ ratio. This can occur by enhanced chemical reduction of $NAD^+$ to NADH mediated by the $Ca^{2+}$-dependent facilitation of TCA cycle dehydrogenases[8,14–16]. Measurements of activity dependent changes in NAD(P)H autofluorescence have been extensively used to monitor adaptive energy metabolism in brain tissue, but the importance of the MCU to these processes was not certain[73,82,83]. For example, evoked NAD(P)H transients in the

hippocampus have been shown to either be independent of or only partially dependent on $Ca^{2+}$ influx from the extracellular space[82,83]. Therefore, the present work clarifies the importance of mitochondrial $Ca^{2+}$ entry via the MCU to metabolic adaptation in activated pyramidal neurons in situ. Our findings in cortical neurons are consistent with recent work in the hippocampal dentate gyrus, which showed that MCU knockdown partially disrupted the evoked mitochondrial NAD(P)H dip and strongly attenuated the delayed NAD(P)H overshoot phase[75].

In cultured neurons, direct measurements of presynaptic ATP revealed that mitochondrial $Ca^{2+}$ uptake was required to sustain ATP levels during activity, and thereby maintained synaptic vesicle endocytosis[2]. An analogous process likely occurs in the somatodendritic compartment in situ, as the NADH overshoot promoted by the MCU would be expected to contribute to oxidative phosphorylation[14]. As we found that substantial mitochondrial $Ca^{2+}$ uptake only occurs during enhanced action potential firing rate, the MCU appears to serve as a conditional regulator of energy metabolism during periods of intense energy demand. This mirrors findings in cardiac tissue, where mitochondrial $Ca^{2+}$ uptake is required for producing sufficient ATP to sustain action potential firing selectively during fight-or-flight accelerations in heart rate[84]. The functional consequence of MCU-dependent ATP production in the somatodendritic compartment is unclear. Enhanced mitochondrial metabolic activity could contribute to the production of ATP required for maintaining membrane ion gradients[1,85] or supporting activity-dependent plasticity in response to high-frequency signalling[86]. Aside from controlling neuroenergetic adaptation, changes in mitochondrial $Ca^{2+}$ evoked by high-frequency firing could also alter the production of reactive oxygen species or mitochondrial metabolites, which can serve as cellular signalling molecules[7,87].

Our results provided evidence that the degree of coupling between neuronal activity and mitochondrial $Ca^{2+}$ uptake differs between discrete brain regions. The greater coupling between spike firing and MCU activation in hippocampal neurons relative to cortical neurons can be accounted for by various factors. One explanation is cell type-dependent variation in the properties of evoked cytosolic $Ca^{2+}$ signals. For instance, discrete neuronal populations exhibit unique $Ca^{2+}$ buffering capacities[88] and the relative abundance of voltage-gated $Ca^{2+}$ channel types can also vary, with some channel classes showing differential efficacy in triggering $Ca^{2+}$ uptake by the mitochondria[21,88–91]. The differential coupling could also reflect regional differences in intrinsic mitochondrial properties, such as energy metabolism[42,44] or the function of the MCU. For example, although the MCU complex is ubiquitously expressed, emerging evidence suggests that mitochondrial $Ca^{2+}$ uptake in non-neuronal cells may vary substantially as a result of post-translational modification[92,93] or changes in the relative expression of the MCU and the MICU regulatory subunits[41,43,94–96]. Interestingly, an analogous process may occur within the nervous system, as the expression profile of the MCU complex components is different between neurons and astrocytes as well as hippocampal subregions[41,94]. Our work provides evidence that is consistent with such variation in the function of the MCU complex between brain regions. Physiologically, the brain region-dependent variation in coupling between activity and mitochondrial $Ca^{2+}$ uptake may represent a mechanism for cell-specific adaption of energy generation. This may be advantageous, as discrete neuronal populations can exhibit distinct energy demands during signalling[97]. In addition to its physiological role, excessive mitochondrial $Ca^{2+}$ uptake can also contribute to metabolic dysfunction, reactive oxygen species generation, and neuronal death[27,98]. Therefore, heightened mitochondrial $Ca^{2+}$ uptake in the hippocampal neurons could contribute to the sensitivity of this region to dysfunction in

various pathologies, such as Alzheimer's disease or ischemia[99], in which mitochondrial dysfunction and $Ca^{2+}$ overload have been implicated[25,27].

Prior studies have demonstrated that $Ca^{2+}$ buffering by the mitochondria clamps cytosolic $Ca^{2+}$ concentration and thereby shapes aspects of synaptic transmitter release in central synapses[22,23,100]. However, its functional relevance in other subcellular compartments of central neurons during signalling has not been well established[100,101]. Our work demonstrates a potent effect of mitochondrial $Ca^{2+}$ uptake on cytosolic $Ca^{2+}$ levels in the somatodendritic compartment, selectively during high-frequency action potential firing. By doing so, mitochondrial $Ca^{2+}$ uptake strongly reduced the cytosolic $Ca^{2+}$ available to activate the $K^+$ current underlying the sAHP. Aside from the canonical $Ca^{2+}$-dependent sAHP, disruption of intracellular ATP production by MCU blockade could conceivably lead to the activation of a $K_{ATP}$ channel-mediated sAHP[102]. Nevertheless, this seems unlikely to have contributed to our results considering that increasing intracellular $Ca^{2+}$ buffering prevented the enhancement of the sAHP seen during MCU disruption. The interplay between mitochondrial $Ca^{2+}$ transport, ion channel modulation, and excitability has been demonstrated in olfactory sensory neurons, myenteric neurons, invertebrate neurons, and non-neuronal cells[103–106]. However, it has not been established whether this link between mitochondrial $Ca^{2+}$ uptake and excitability also occurs in central neurons. Our results reveal that $Ca^{2+}$ buffering by the mitochondria exerts control over neuronal excitability in the brain. By reducing the coupling between $Ca^{2+}$ influx and the activation of the sAHP, mitochondrial $Ca^{2+}$ uptake is poised to promote neuronal signalling by disrupting negative feedback control of spike firing and synaptic integration[46,107,108]. Such changes in the coupling between spike firing and activation of the sAHP can have meaningful consequences for information processing, as seen in the hippocampus during age-related learning disruption[109,110].

Collectively, our in situ study demonstrates that mitochondrial $Ca^{2+}$ uptake in pyramidal neurons is tuned to enhanced action potential firing rate and promotes neuronal activity through a dual role in metabolic regulation and excitability control (Fig. 8). This dual role of mitochondrial $Ca^{2+}$ uptake in the somatodendritic compartment could shape action potential output as well as synaptic integration and plasticity in the dendrites, where mitochondrial $Ca^{2+}$ uptake was recently shown to occur during coincident pre- and post-synaptic activity[79]. We expect that MCU activation in pyramidal neurons influences sensorimotor processing and other cognitive functions, as in vivo recordings of cortical and hippocampal pyramidal neurons have shown firing intensities within the range of our experimental stimuli[39,62,111,112]. For example, CA1 pyramidal neurons transition from states of minimal action potential firing (<2 Hz) to periods of high frequency action potential bursts (>50 Hz) occurring at theta frequency, and thereby act as place cells during exploratory behaviour[62,63]. Also, in the visual cortex, pyramidal neurons have little spontaneous activity (<1 Hz), but fire at ~10-20 Hz in response to the application of a visual stimulus[39]. Consistent with MCU activation during information processing in the brain, a recent in vivo study demonstrated that mitochondrial $Ca^{2+}$ elevations occur in neurons of the motor and visual cortex in response to motor behaviour or visual stimulation, respectively[37]. Our results suggest that MCU activation may be engaged during such brain activity to sustain neuronal signalling through its influence on energy metabolism and excitability. Consistent with this, preventing mitochondrial $Ca^{2+}$ uptake by the MCU has been found to disrupt spike firing rate increases in cortical neurons during sensory stimulation[113]. Moreover, the activation of fast neuronal network oscillations, which are associated with a multitude of cognitive processes, require high mitochondrial performance, and are prevented by MCU knockdown in excitatory neurons[4,36]. A similar property may also be important in fast-spiking GABAergic interneurons, as these cells exhibit comparatively higher frequency firing and more prolonged action potential firing[114]. As activation of the MCU is primarily engaged during high-frequency signalling, this pathway could also serve as a target for controlling excessive neuronal activity in various brain pathologies.

## Methods

**Animals**. Sprague Dawley rats were used for this study and were purchased from Charles River Laboratories. Rat housing and experimental procedures were performed in accordance with Canadian Council on Animal Care (CCAC) regulations and protocols approved by the University of British Columbia Animal Care Committee. Animals were group housed, fed ad libitum, and kept on a 12/12 hr light/day cycle. Rats were 17-28 days old when experiments were conducted.

**Brain slice preparation**. Rats were anesthetized with isoflurane in an induction chamber and subsequently decapitated. The brains were then quickly removed and placed in ice-cold slicing solution containing the following (in mM): 120 N-methyl-D-glucamine, 2.5 KCl, 25 NaHCO$_3$, 1 CaCl$_2$, 7 MgCl$_2$, 1.25 NaH$_2$PO$_4$, 20 D-glucose, 2.4 sodium pyruvate, and 1.3 sodium L-ascorbate. The slicing solution was continuously bubbled with 95% O$_2$ and 5% CO$_2$. Brains were then cut with a vibratome (Leica VT1200S) into coronal or horizontal slices (400 μm thick) for cortical and hippocampal experiments, respectively. Brain slices were immediately transferred to a chamber containing artificial cerebral spinal fluid (aCSF), which was warmed to 32 °C and continuously oxygenated with 95% O$_2$ and 5% CO$_2$. aCSF contained (in mM): 126 NaCl, 2.5 KCl, 26 NaHCO$_3$, 2 CaCl$_2$, 1.5 MgCl$_2$, 1.25 NaH$_2$PO$_4$, and 10 D-glucose, pH 7.3–7.4, osmolarity 310 mOsm. A Ca$^{2+}$-free aCSF solution was used to test the involvement of extracellular Ca$^{2+}$ entry on mitochondrial Ca$^{2+}$ signals, and contained (in mM): 126 NaCl, 2.5 KCl, 26 NaHCO$_3$, 3.5 MgCl$_2$, 1.25 NaH$_2$PO$_4$, 10 D-glucose, 1 EGTA.

**AAV injections**. For AAV injections, neonatal rats, 1-3 days old, were anesthetized with isoflurane and placed in a stereotaxic frame. A glass micropipette attached to a Hamilton Syringe was then used to pierce the skull unilaterally at a region ~2/3 the distance between lambda and the eye. AAV solution (2 uL total; titer: $1.0 \times 10^{13}$-$1.5 \times 10^{13}$ genome copies/mL) was then injected into the cortex at a rate of 600 nL/min using a motorized stereotaxic injector (Stoelting). Following injections, animals recovered from anesthesia and were then returned to the home cage. After 2-4 weeks, animals were sacrificed to prepare acute brain slices for experiments. AAVs were purchased from the Canadian Neurophotonics Platform Viral Vector Core Facility and Addgene: AAV9-synapsin-GCAMP6f (Addgene, 100837-AAV9); AAV2/9-synapsin-mitoRGECO1.0 (Neurophotonics); AAV2/9-synapsin-mito-LaRGECO1.2 (Neurophotonics); AAV2/9-synapsin-mitoGFP (Neurophotonics).

**Chemicals and reagents**. Drugs and dyes used for experiments are listed in final concentration (in mM): 0.02 Ru360 (Sigma Aldrich, 557440), 0.01 rotenone (Sigma Aldrich, R8875), 0.05 Alexa 488 hydrazide (Thermo Fisher Scientific, A10436), 0.05 Alexa 594 hydrazide (Thermo Fisher Scientific, A10438), 0.4 Fluo5N pentapotassium salt (Thermo Fisher Scientific, F14203). All drugs or chemicals were dissolved into the internal solution of the patch pipette or the aCSF. DMSO was used as the vehicle for rotenone and was diluted to a maximal working concentration of 0.1%.

**Two-photon imaging in acute brain slices**. Live brain slice imaging was performed using a two-photon laser-scanning microscope (LSM 7MP, Zeiss) equipped with a Zeiss water immersion objective lens (Plan-Apochromat 20×/numerical aperture 1.0). Excitation light was provided by a mode-locked Ti:Sapphire laser (Chameleon Ultra II, Coherent), which was pulsed at 80 MHz (140 fs per pulse). The excitation wavelength for each experiment was adjusted according to the fluorescent probes being imaged: mitoGFP & mRGECO, 1000 nm; NAD(P)H, 750 nm; Fluo5N & Alexa 594, 800 nm; mRGECO & Alexa 488, 1040 nm; mLaRGECO & Alexa 488, 1040 nm; mRGECO and GCAMP6f, 1040 nm. Emitted light was first passed through a 700 nm shortpass IR filter and then split with a 560 nm long-pass dichroic mirror (Chroma). Emitted light was then detected by two separate LSM BiG GaAsP detectors after passing through the appropriate emission filters (Chroma): green filter (520/60 nm) for MitoGFP, GCAMP6f, Fluo5N, Alexa488; blue filter for NAD(P)H (460/50 nm), red filter (630/75 nm) for mRGECO, mLaRGECO, Alexa594. Images were acquired using 8-16 line averaging and were processed and stored with Zen software (Zeiss).

**Patch-clamp electrophysiology**. Following brain slice preparation and a 30-min recovery period, tissue was transferred to the microscope recording chamber and perfused with aCSF at 2–3 mL/min. aCSF was bubbled with 95% O$_2$/5% CO$_2$ and

warmed to 32 °C with a stage heater (Luigs & Neumann). Pyramidal neurons in the cortex (layer 3-5) and hippocampus (CA1 and CA3) were identified with transmitted light optics and an IR-1000 (DAGE-MTI) camera mounted to the LSM 7MP microscope. All neurons used for experiments were located >50 μm below the slice surface and had a resting membrane potential ranging from −55 to −70 mV. Neurons were patch-clamped using glass pipettes pulled from borosilicate glass capillaries (World Precision Instruments) with a P-97 Flaming/Brown Micropipette Puller (Sutter Instrument). Patch pipettes had a resistance of 3-8 MΩ and contained an intracellular solution comprised of the following (in mM): 110 K-Gluconate; 3 KCl; 10 Na-Gluconate; 2 $MgCl_2$; 4 $K_2$-ATP; 0.5 Na-GTP; 10 HEPES; 0.138 $CaCl_2$; 0.4 EGTA; pH adjusted to 7.2 with KOH and osmolarity adjusted to 290 mOsm. For high $Ca^{2+}$ buffering experiments, the internal solution contained (in mM): 70 K-Gluconate; 3 KCl; 10 Na-Gluconate; 2 $MgCl_2$; 4 $K_2$-ATP; 0.5 Na-GTP; 10 HEPES; 4.15 $CaCl_2$; 10 $K_4$-BAPTA; pH adjusted to 7.2 with KOH and osmolarity adjusted to 290 mOsm. The free $Ca^{2+}$ concentration of both internal solutions was calculated with WEBMAXC Standard (UC Davis) to be between 70-170 nM. For some experiments, the internal solution was supplemented with fluorescent dyes or drugs at the following concentrations (in mM): 0.05 Alexa 594 hydrazide; 0.4 $K_5$-Fluo5N; 0.05 Alexa 488 hydrazide; 0.02 Ru360. For all patch-clamp recordings, cells were dialyzed for a minimum of 15 min after whole-cell breakthrough before starting experiments. Neurons were confirmed to be pyramidal cells by morphological inspection, revealed by Alexa488 or Alexa594 fluorescence. Only neurons that possessed a pyramid-shaped cell body, large apical dendrite, and multiple basal dendrites were used in the study. Neuronal membrane voltage and current were recorded and controlled using a MultiClamp 800b amplifier, Digidata 1440 A digitizer, and Clampex 10.2 software (Axon Instruments, Molecular Devices). During voltage-clamp recordings, capacitance transients were compensated, and series resistance (8-20 MΩ) corrected to >40%. During whole-cell current clamp recordings, membrane voltage was maintained between −60 and −65 mV and series resistance was compensated using the bridge balance. Voltage offsets were nulled prior to seal formation and the membrane potentials presented were adjusted to correct for the liquid junction potential. Current and membrane voltage signals were low pass filtered by a built-in Bessel filter at 1 kHz and 3 kHz, respectively and digitized at 10 kHz.

**Data analysis**. All image analysis and image processing was performed using ImageJ (v 1.53). Prior to analysis, images were first processed to correct for translational movement using the correct 3D drift plugin on ImageJ. Activity-evoked fluorescence changes for mRGECO, mLaRGECO, GCAMP6f, and NAD(P)H were measured from a somatic region of interest (ROI) in pyramidal neurons and were defined as $\Delta F/F_o = [(F_1) - (F_o)] / (F_o)$. Where $F_1$ and $F_0$ represent the fluorescence at a given time point and the mean baseline fluorescence, respectively. Background fluorescence subtraction was not performed for these experiments as all regions neighboring the ROI contained meaningful fluorescence signal. Changes in Fluo5N fluorescence were determined similarly, but using background fluorescence subtraction: $\Delta F/F_o = [(F_1 - B_1) - (F_o - B_o)] / (F_o - B_o)$. $B_0$ and $B_1$ represent the background fluorescence during the baseline period and at a given time point during the time series recording, respectively. The ROIs used for measuring cytosolic $Ca^{2+}$ in the soma with Fluo5N were drawn to exclude the nuclear fluorescence signal. For some data sets, fluorescence changes were also measured from ROIs placed over the proximal apical dendrite. Spontaneous mRGECO fluorescence transients were quantified by measuring the number and magnitude of fluorescence responses from a grid of 10 ×10 μm ROIs mapped onto each experiment time series. A spontaneous event was defined as a fluorescence increase in a given ROI that was >3 standard deviations above the mean baseline fluorescence. For analysis, peak or min $\Delta F/F_o$ responses were defined as the maximal changes in fluorescence after the stimulus onset time. The $Ca^{2+}$ rise time was defined as the time from stimulus onset to half of the peak train-evoked $\Delta F/F_o$. The coupling ratio between mitochondrial $Ca^{2+}$ and cytosolic $Ca^{2+}$ was calculated as the ratio of mRGECO $\Delta F/F_o$ and GCAMP $\Delta F/F_o$ at 2-sec post-stimulus onset. The rate of $Ca^{2+}$ recovery in the mitochondria or cytosol was quantified by fitting the recovery phase with a monoexponential decay function or by measuring the time span from the end of the stimulus to the point at which $\Delta F/F_o$ was 50% of the peak fluorescence response. Images presented for publication were minimally spatially filtered and in some cases were also averaged across multiple time points. For some images, a mask of the neuronal outline was produced by forming a binary image from the intracellular Alexa594 fluorescence signal.

The analysis of membrane voltage and the current was performed using Clampfit (v10.2). The sAHP duration was determined by measuring the time span between the end of the action potential train to the point of 50% recovery of the membrane voltage back to the pre-stimulus baseline. The $I_{sAHP}$ magnitude was determined by quantifying the post-depolarization area above prestimulus baseline (ms*pA/pF). This value equates to charge and was measured starting at 30 ms after the end of the step depolarization.

**Statistics and reproducibility**. Statistics were performed using GraphPad Prism (v7). All summary data shown represent the mean ± standard error of mean. For each data set, the cell replicates and the animal replicates are described within the summary data and figure legends. The cell replicate number is represented by the 'n' value and the animal replicate number by the 'N' value. Unless otherwise stated, all presented summary data represent distinct samples. Outliers were identified using the ROUT test and removed prior to further statistical testing. The Shapiro-Wilk normality test was used to examine data sets for normality. If data were normal, a Student's paired or unpaired t test was used to test for differences between two means, while a one-way ANOVA with a Dunnett's multiple comparisons test was used to test for differences between multiple means and a control condition. If the data were not normally distributed, a Mann–Whitney U test was used for two means, whereas a Friedman test with Dunn's multiple comparison test was used to compare multiple means to a control condition. A Welch's t test was used to compare two data sets if the variances differed significantly. Tests were considered statistically significantly different at $p \leq 0.05$. The exact p-values are indicated in the figure legends.

**Reporting summary**. Further information on research design is available in the Nature Research Reporting Summary linked to this article.

## Data availability

Source data used for summary figures are provided in the Supplementary Data 1 file. AAVs used in this study are available from the Canadian Neurophotonics Platform Viral Vector Core Facility and Addgene: AAV9-synapsin-GCAMP6f (Addgene, 100837-AAV9); AAV2/9-synapsin-mitoRGECO1.0 (Neurophotonics); AAV2/9-synapsin-mitoLaRGECO1.2 (Neurophotonics); AAV2/9-synapsin-mitoGFP (Neurophotonics). Raw data from the current study is available from the corresponding author on reasonable request.

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

## Acknowledgements

This work was supported by a grant from the Canadian Institutes of Health Research (FDN-148397 to B.A.M.). C.J.G was supported by postdoctoral fellowships from the Canadian Institutes of Health Research and the Michael Smith Foundation for Health Research. The authors would like to thank Drs. Nicholas Weilinger, Louis-Philippe Bernier, and Stefan Wendt for their insightful feedback on the manuscript draft. We would also like to thank Rayshad Gopaul and Lucy Yang for their excellent technical assistance.

## Author contributions

C.J.G. and B.A.M. conceived and designed the study. C.J.G. performed all experiments and data analysis. C.J.G and B.A.M. wrote and edited the manuscript.

## Competing interests

The authors declare no competing interests.

## Additional information

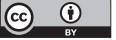

