## [Peer Review File · Communications Biology]

Reviewers' comments:

Reviewer #1 (Remarks to the Author):

Groten and MacVicar (COMMSBIO-22-0355-T) explored the role of the mitochondrial calcium uniporter (MCU) in neurometabolic coupling and neuronal excitability in neocortical and hippocampal slices of the mouse. They used patch-clamp recordings and two-photon imaging of mitochondrial calcium, cytosolic calcium, and NAD(P)H with high spatiotemporal resolution in individual pyramidal cells. The authors provide substantial evidence that the MCU is engaged by accelerated spiking to facilitate neuronal activity through simultaneous control of energy metabolism and neuronal excitability. The topic is timely, and the novel findings are based on a large series of carefully planned and executed experiments, including appropriate controls. The figures are clear, and the manuscript is well written.

The authors might consider the following criticism to improve clarity of the manuscript.

Major points:

1. Results and M+M. The criteria for inclusion (exclusion) of the investigated neurons ('visually identified pyramidal neurons') are somewhat unclear. Did the authors check the electrophysiological properties of the neurons? Please, specify.
2. Results and Discussion. a) The statements about action potential frequencies in hippocampal and neocortical pyramidal cells are somewhat confusing. Examples: 'typically have low spike firing rates' (line 152) versus 'firing rate ranging from 5 to 100 Hz' (line 430). This needs to be more carefully phrased and referenced with reports on spiking rates in pyramidal cells during network oscillations (or behavioural tasks) and, perhaps, versus fast-spiking interneurons. b) Similarly, it is unclear to which natural neuronal network state(s) 'high frequency bursts applied at theta frequency' (line 189) might correspond (theta-gamma oscillations?). Please, check and revise.
3. Results and Discussion. Under physiological conditions, action potentials are generated in the axon initial segment and can actively backpropagate into the dendrites of many neuronal subtypes. Do the authors expect similar levels of backpropagation during experimental stimulation (train of depolarizing current pulses at 50 Hz for 4 seconds, line 113) of pyramidal cells in the whole-cell patch-clamp configuration? Please, discuss.
4. Results (Figure 1C). '(...), with each pulse consistently evoking a single action potential' (line 114). This is not clear from the electrophysiological traces at this scaling. Please, revise.
5. Results (Figure 4A-D). It seems that the ROI was set to the entire soma of pyramidal cells. What was the rationale to include the nucleus that is free of mitochondria? Do the authors obtain similar results when setting a somatic ROI (or various smaller ROIs) adjacent to the nucleus?

Minor points:

6. Abstract and elsewhere. 'NADH generation' (line 30). Maybe better use 'oxidation of NAD(P)H' and 'reduction of NAD(P)⁺'.
7. Results and Legends. The sample sizes are somewhat unclear. Please, specify 'n' (e.g., neurons) from 'N' (mice) in each legend to highlight the biological robustness of the data sets.
8. Results. 'This NAD(P)H overshoot is consistent with a prolonged increase in NADH generation by the mitochondria' (line 229). What about glycolysis? Please, discuss.
9. Discussion. Use of subheadings would facilitate reading of the Discussion. Please, revise.

Reviewer #2 (Remarks to the Author):

The manuscript by Groten and MacVicar reports the relationship between action potential firing rate and MCU-mediated calcium uptake and how this potentially influences both energy production and the extent of action potential afterhyperpolarization. These findings are significant since they go toward explaining how neurons maintain/increase energy production for sustained neuronal activity. Both readers interested generally in calcium signaling and specifically in neuronal calcium signaling will find this research relevant. The manuscript is nicely written and does an excellent job citing publications of other researchers in this field. The data are high quality and presented clearly. The statistical analyses conducted are explained clearly in the methods section.

One minor comment is that on line 192 the units are missing in the text for 50% recovery time.

Overall this is a very nicely prepared and interesting manuscript.

Reviewer #3 (Remarks to the Author):

This paper demonstrated the importance of MCU-mediated mitochondrial Ca^{2+} uptake to the regulation of neuronal bioenergetics and excitability in the brain. Their approach enlisted single-cell patch-clamp recording, two-photon imaging of the mitochondrial and cytosolic Ca^{2+} as well as endogenous NAD(P)H in acute brain slices. Evidence for the involvement of MCU was obtained with patch pipette dialysis of Ru360, a MCU inhibitor. Their main findings include the following. (1) MCU-mitochondrial Ca^{2+} uptake is tuned by spike frequency with a threshold, and the cytosolic-to-mitochondrial Ca^{2+} coupling varies quantitatively between brain regions (the cortex versus the hippocampus); (2) MCU-mediated mitochondrial Ca^{2+} uptake mitigates spike-induced cytosolic Ca^{2+} , and retrospectively affects spike frequency and excitability (slow afterhyperpolarization), probably by repressing Ca^{2+} -dependent K^{+} currents; (3) MCU-mitochondrial Ca^{2+} uptake exerts a biphasic change on NAD(P)H autofluorescence, suggestive of a boost of both mitochondrial respiration and TCA metabolism.

Overall this is an interesting and solid study on important topics, though some conclusions are confirmatory. The experiments were well-executed, and the manuscript is well-written. I have only a few comments as the following.

In the establishment of MCU as the principal route for mitochondrial Ca^{2+} uptake and ensuing signaling events within and beyond the organelle, the authors heavily relied on the single chemical agent, Ru360. Given the concern on promiscuous actions of this agent in cellular context, I am wondering whether they could adopt independent approaches, e.g., the use of MCU-knockout models?

In addition, the authors could briefly discuss about a highly relevant work by Stoler et al, which appeared recently in eLife (Stoler et al., 2022).

Response to the Reviewers:

Reviewer 1:

“Groten and MacVicar (COMMSBIO-22-0355-T) explored the role of the mitochondrial calcium uniporter (MCU) in neurometabolic coupling and neuronal excitability in neocortical and hippocampal slices of the mouse. They used patch-clamp recordings and two-photon imaging of mitochondrial calcium, cytosolic calcium, and NAD(P)H with high spatiotemporal resolution in individual pyramidal cells. The authors provide substantial evidence that the MCU is engaged by accelerated spiking to facilitate neuronal activity through simultaneous control of energy metabolism and neuronal excitability. The topic is timely, and the novel findings are based on a large series of carefully planned and executed experiments, including appropriate controls. The figures are clear, and the manuscript is well written. The authors might consider the following criticism to improve clarity of the manuscript.”

Comment 1:

“Results and M+M. The criteria for inclusion (exclusion) of the investigated neurons ('visually identified pyramidal neurons') are somewhat unclear. Did the authors check the electrophysiological properties of the neurons? Please, specify.”

Response to Comment 1:

The reviewer would like clarification on the criteria that we used to determine whether a recorded neuron was, or was not, a pyramidal neuron. Cell type determination was achieved primarily by inspecting cell morphology, since patch-clamp experiments were performed in combination with two-photon microscopy. Cell morphology was determined using visualization of cell fluorescence derived from morphology dyes (e.g. Alexa 594 or Alexa 488) applied intracellularly via the patch pipette. Neurons were confirmed to be pyramidal cells by the presence of a large apical dendrite, pyramid shaped cell body, and multiple basal dendrites. Regarding electrophysiology properties, we did not systematically evaluate the spike firing properties of each neuron to differentiate pyramidal cells from non-pyramidal cells. However, our recordings clearly demonstrate the presence of an activity-evoked slow afterhyperpolarization (see Figure 6 & 7) which is absent from fast spiking interneurons^[1]. On rare occasions, neurons were patch-clamped that possessed a stellate morphology, consistent with a non-pyramidal cell. These neurons were excluded from our data sets. We have now provided a more detailed description of our method for determining the inclusion of neurons in our data.

Comment 1 text changes:

Lines 110-114 (Results): “To monitor mitochondrial Ca²⁺ dynamics in relation to evoked neuronal activity, we performed whole-cell patch-clamp recordings from layer 3-5 pyramidal neurons expressing mRGECO. Cells were filled with Alexa-488 and confirmed to be pyramidal

neurons based on the presence of a pyramid shaped cell body, large apical dendrite, and multiple basal dendrites (Fig. 1C).”

Lines 540-543 (Materials & Methods): “Neurons were confirmed to be pyramidal cells by morphological inspection, revealed by Alexa488 or Alexa594 fluorescence. Only neurons that possessed a pyramid shaped cell body, large apical dendrite, and multiple basal dendrites were used in the study.”

Comment 2:

“Results and Discussion. a) The statements about action potential frequencies in hippocampal and neocortical pyramidal cells are somewhat confusing. Examples: 'typically have low spike firing rates' (line 152) versus 'firing rate ranging from 5 to 100 Hz' (line 430). This needs to be more carefully phrased and referenced with reports on spiking rates in pyramidal cells during network oscillations (or behavioural tasks) and, perhaps, versus fast-spiking interneurons. b) Similarly, it is unclear to which natural neuronal network state(s) 'high frequency bursts applied at theta frequency' (line 189) might correspond (theta-gamma oscillations?). Please, check and revise.”

Response to Comment 2:

The reviewer suggests that we more clearly state the known spike firing rates of pyramidal neurons *in vivo* in relation to behavioural tasks or neuronal network activity, as well as interneuron firing rates. Numerous *in vivo* studies recording from single pyramidal neurons within the cortex or hippocampus have shown comparatively low levels of action potential firing under most conditions (~0-5 Hz) [2-5]. However, these cells rapidly increase their firing rate following circuit activation during sensory processing or specific cognitive tasks. For example, in the cat visual cortex, pyramidal neurons have little spontaneous activity (<1 Hz), but fire at ~10-20 Hz in response to the application of a visual stimulus [2]. Similarly, *in vivo* recordings from macaque prefrontal cortex pyramidal neurons show that firing rate shifts from a low basal level (<5 Hz) to a firing rate of ~10-30 Hz for several seconds during working memory tasks [3]. In rats or mice, CA1 pyramidal neurons function as place cells during exploratory behaviour and become most active when the animal is in a discrete physical location-the cell's place field [4, 5]. During place cell activation, these neurons transition from states of minimal action potential firing (<2 Hz) to periods of high frequency activity, often in the form of action potential bursts [4, 5]. These bursts occur at theta frequency (~5 Hz) and consist of a cluster of several high frequency action potentials (>50 Hz) [4, 5]. With respect to fast spiking inhibitory interneurons, these cells typically fire at higher frequencies than pyramidal neurons during the same brain activity states [6, 7]. Collectively, we expect that *in vivo* cortical and hippocampal pyramidal neuron firing intensity is sufficiently similar to our experimental stimuli to produce MCU activation and mitochondrial Ca²⁺ uptake during brain activity. Consistent with this, brain activation during sensory or motor functions has been shown to trigger mitochondrial Ca²⁺ uptake in cortical pyramidal neurons *in vivo* [8]. We adjusted our phrasing and expanded on these points throughout the manuscript.

The reviewer also mentions that it is not clear how the spike firing (e.g. theta burst protocol) experimentally evoked in pyramidal neurons relates to natural brain network states. As mentioned above, the firing pattern used for our protocol (4 action potentials at 50 Hz, applied at theta frequency [5 Hz]) is like that recorded *in vivo* from single neurons during exploratory behaviour in rodents [4, 5]. This theta burst firing of single CA1 place cells during exploratory behaviour occurs alongside network theta oscillations in the hippocampus [5, 9]. The temporal relationship between the oscillation phases of single neuron bursting and network theta rhythms shifts during animal movement through a place field and may therefore provide a representation of animal spatial location [9, 10]. We have attempted to clarify the relevance of the theta firing protocol in the revised manuscript.

Comment 2 text changes:

Line 156-158: “*In vivo*, pyramidal neurons in the cortex and hippocampus typically have low spike firing rates (<5 Hz) but can enter states of high frequency action potential firing (10-50 Hz) during sensory processing or other cognitive tasks [2-5]”

Line 188-192: “We also assessed whether mitochondrial Ca^{2+} uptake occurred in response to activity patterns that are relevant to hippocampal neuron activity *in vivo*. To do this, we imaged CA1 pyramidal neurons while triggering theta burst firing. This stimulus pattern mimics the high frequency action potential bursts (>50 Hz) that occur at theta frequency (~5 Hz) in CA1 neurons during rodent exploratory behaviour and learning processes [4, 5, 10, 11]”

Line 444-451: “We expect that MCU activation in pyramidal neurons influences sensorimotor processing and other cognitive functions, as *in vivo* recordings of cortical and hippocampal pyramidal neurons have shown firing intensities within the range of our experimental stimuli [2, 5, 12, 13]. For example, CA1 pyramidal neurons transition from states of minimal action potential firing (<2 Hz) to periods of high frequency action potential bursts (>50 Hz) occurring at theta frequency, and thereby act as place cells during exploratory behaviour [4, 5]. Also, in the visual cortex, pyramidal neurons have little spontaneous activity (<1 Hz), but fire at ~10-20 Hz in response to the application of a visual stimulus [2].”

Line 460-462: “A similar property may also be important in fast spiking GABAergic interneurons, as these cells exhibit comparatively higher frequency firing and more prolonged action potential firing [14]”

Comment 3:

“Results and Discussion. Under physiological conditions, action potentials are generated in the axon initial segment and can actively backpropagate into the dendrites of many neuronal subtypes. Do the authors expect similar levels of backpropagation during experimental stimulation (train of depolarizing current pulses at 50 Hz for 4 seconds, line 113) of pyramidal cells in the whole-cell patch-clamp configuration? Please, discuss.”

Response to comment 3:

The reviewer is interested in whether we expect action potentials to backpropagate into the dendrites during the stimulus paradigm used in our experiments. Several studies using dendritic patch-clamp recordings and Ca^{2+} imaging in pyramidal neurons have shown that triggering action potential trains at the soma results in backpropagating action potentials which can elevate dendritic intracellular Ca^{2+} [15, 16]. However, dendritic action potential magnitude attenuates with ongoing stimulation, meaning that action potentials become progressively less effective at propagating to distant dendritic sites and initiating Ca^{2+} influx [15, 16]. The relationship between spike train firing frequency and the effectiveness of backpropagating action potentials has also been examined in pyramidal neurons. Progressively greater train firing frequencies (ranging from 5 to ~65 Hz) results in more attenuation of backpropagating action potential magnitude [16]. Our 50 Hz, 4 second train is both high frequency and contains a series of action potentials. Therefore, we expect that this stimulus elicits backpropagating action potentials into the dendrites, but which are strongly attenuated in magnitude over time and with distance from the soma. As a result, we anticipate that the evoked cytosolic and mitochondrial Ca^{2+} responses during this stimulus are constrained to the soma and proximal dendrites. Figure 2C shows a representative example of a 50-Hz, 4-sec train that evokes a cytosolic and mitochondrial Ca^{2+} elevation in the proximal apical dendrite, likely because of backpropagating action potentials. In the revised manuscript, we now refer to backpropagating action potentials as the putative mechanism causing the train-evoked cytosolic and mitochondrial Ca^{2+} in the dendrites.

Comment 3 text changes:

Line 139-144: “Removal of extracellular Ca^{2+} from the ACSF completely eliminated spike-evoked mitochondrial Ca^{2+} uptake (Fig. 2A), suggesting that the source of cytosolic Ca^{2+} driving the mitochondrial Ca^{2+} elevation is plasma membrane Ca^{2+} influx, rather than intracellular Ca^{2+} release. This Ca^{2+} influx is likely mediated by the activation of voltage-gated Ca^{2+} channels in the soma, as well as the dendrites through backpropagating action potentials [15-17].”

Comment 4:

“Results (Figure 1C). '(...), with each pulse consistently evoking a single action potential' (line 114). This is not clear from the electrophysiological traces at this scaling. Please, revise.”

Response to Comment 4:

We have changed our representative trace in Figure 1C. It now shows that each individual current pulse triggers a single action potential in a 50 Hz train stimulus. The updated Figure is shown on the next page.

Figure 1

Comment 5:

“Results (Figure 4A-D). It seems that the ROI was set to the entire soma of pyramidal cells. What was the rationale to include the nucleus that is free of mitochondria? Do the authors obtain similar results when setting a somatic ROI (or various smaller ROIs) adjacent to the nucleus?”

Response to Comment 5:

This is a fair point. The ROI placement around the entirety of the cell body was a simplified method for measuring somatic NAD(P)H autofluorescence. To assess whether this affected our measured NAD(P)H transients, we evaluated data from Figure 4C. We compared fluorescence measurements from 6 cells acquired using a whole somatic ROI (ROI including nucleus), with data collected from the same images, but with somatic ROIs excluding the nucleus (ROI excluding nucleus). As shown in the figure below, the magnitude and temporal dynamics of the NAD(P)H transients evoked by a 50 Hz train were not different between somatic ROIs including the nucleus vs excluding the nucleus. This lack of effect is consistent with the absence of autofluorescence in the nucleus (see Figure 4B,D). Therefore, we do not anticipate that the use of a whole soma ROI adversely impacted our results and conclusions.

Comment 6:

“Abstract and elsewhere. 'NADH generation' (line 30). Maybe better use 'oxidation of NAD(P)H' and 'reduction of NAD(P)+'.”

Response to Comment 6:

The reviewer suggests that we replace NADH or NAD with NAD(P)H or NAD(P)+. NAD(P)H refers specifically to the measured tissue autofluorescence and is so named because of the inability to discern the fluorescence of NADH from NADPH. Thus, NAD(P)H has a meaning that is similar to, but distinct from NADH. We therefore refer to measured tissue autofluorescence in the manuscript as NAD(P)H, whereas we used NADH/NAD to refer

specifically to the coenzymes, not the autofluorescence per se. As the reviewer suggested, we modified the text to use the term ‘oxidation of NADH’ or ‘reduction of NAD⁺’ where possible.

Comment 6 text changes:

Line 30: “chemical reduction of NAD⁺ to NADH”

Line 234: “chemical reduction of NAD⁺ to NADH⁺”

Line 256-257: “These data indicate that the MCU has a prominent role in modulating NADH oxidation and the reduction of NAD⁺ to NADH during high frequency action potential firing”

Line 360: “chemical reduction of NAD⁺ to NADH”

Comment 7:

“Results and Legends. The sample sizes are somewhat unclear. Please, specify 'n' (e.g., neurons) from 'N' (mice) in each legend to highlight the biological robustness of the data sets.”

Response to Comment 7:

As the reviewer suggested, we specified the ‘n’ (neuron count) and ‘N’ (animal count) in each legend. Changes were made and highlighted in the results section, figure legends, and materials and methods.

Comment 7 Text Change:

Line 590-592 (Materials and Methods): “For each data set, the cell replicates and the animal replicates are described within the summary data or figure legends. The cell replicate number is represented by the ‘n’ value and the animal replicate number by the ‘N’ value.”

Line 629-630 (Figure Legend 1): “The number of cell replicates is shown in the box plots as well as the ‘n’ value in the text. The number of animal replicates is represented by the ‘N’ value”

Comment 8:

“Results. 'This NAD(P)H overshoot is consistent with a prolonged increase in NADH generation by the mitochondria' (line 229). What about glycolysis? Please, discuss.”

Response to Comment 8:

The reviewer asks us to discuss the potential contribution of glycolysis to the NAD(P)H overshoot phase following action potential firing. Cytosolic NADH can be generated from the reduction of NAD⁺ during glycolysis, through the action of glyceraldehyde 3-phosphate dehydrogenase (GAPDH) ^[18]. Enhanced glycolysis has been shown to occur in neurons following synaptic activation, thereby supplying pyruvate to the mitochondria and potentially contributing to the maintenance of cytosolic ATP concentration ^[19]. While glycolysis is likely occurring in neurons during our stimulus paradigms, our data suggests that glycolytically produced NADH did not substantially contribute to the measured NADH autofluorescence

overshoot. We found that disrupting either mitochondrial Ca^{2+} uptake by the MCU or mitochondrial complex I largely prevented the overshoot phase (Figure 5). Similar results have been reported in other studies from neurons *in situ* [20] and in culture [21, 22]. The primarily mitochondrial contribution to the NAD(P)H overshoot is likely attributable to NADH having a higher fluorescence in the mitochondrial compartment than in the cytosol, due to its interaction with mitochondrial proteins [23, 24]. We have referred to the potential contribution of NADH production from glycolysis in the revised manuscript.

Comment 8 text changes:

Line 214-216: “NADH is a coenzyme that is generated from the reduction of non-fluorescent NAD^+ by the mitochondrial TCA cycle or via glycolysis in the cytosol [24]”

Line 234-235: “This NAD(P)H overshoot is consistent with a prolonged chemical reduction of NAD^+ to NADH, which can result from mitochondrial TCA cycle activity or glycolysis in the cytosol [19, 20]”

Comment 9:

“Discussion. Use of subheadings would facilitate reading of the Discussion. Please, revise.”

Response to Comment 9:

We have now added subheadings to the discussion.

Comment 9 text changes:

Line 325: “MCU activation and mitochondrial Ca^{2+} uptake is tuned to spike firing frequency”

Line 348-349: “MCU activation facilitates adaptive mitochondrial energy metabolism during high frequency activity”

Line 386-387: “The coupling between action potential firing and mitochondrial Ca^{2+} uptake varies between brain regions.”

Line 413-414: “Mitochondria buffer cytoplasmic Ca^{2+} and modulate the excitability of pyramidal neurons during high frequency action potential firing.”

Line 437: “Functional implications of MCU activation and relevance to neuronal activity in vivo.”

Reviewer 2:

“The manuscript by Groten and MacVicar reports the relationship between action potential firing rate and MCU-mediated calcium uptake and how this potentially influences both energy production and the extent of action potential afterhyperpolarization. These findings are significant since they go toward explaining how neurons maintain/increase energy production for sustained neuronal activity. Both readers interested generally in calcium signaling and specifically in neuronal calcium signaling will find this research relevant. The manuscript is nicely written and does an excellent job citing publications of other researchers in this field. The data are high quality and presented clearly. The statistical analyses conducted are explained clearly in the methods section.”

Comment 1:

“One minor comment is that on line 192 the units are missing in the text for 50% recovery time.”

Response to Comment 1:

We have corrected the text to include the time units.

Comment 1 text changes:

Line 196: “time (sec) to 50% recovery”

Reviewer 3:

“This paper demonstrated the importance of MCU-mediated mitochondrial Ca^{2+} uptake to the regulation of neuronal bioenergetics and excitability in the brain. Their approach enlisted single-cell patch-clamp recording, two-photon imaging of the mitochondrial and cytosolic Ca^{2+} as well as endogenous NAD(P)H in acute brain slices. Evidence for the involvement of MCU was obtained with patch pipette dialysis of Ru360, a MCU inhibitor. Their main findings include the following. (1) MCU-mitochondrial Ca^{2+} uptake is tuned by spike frequency with a threshold, and the cytosolic-to-mitochondrial Ca^{2+} coupling varies quantitatively between brain regions (the cortex versus the hippocampus); (2) MCU-mediated mitochondrial Ca^{2+} uptake mitigates spike-induced cytosolic Ca^{2+} , and retrospectively affects spike frequency and excitability (slow afterhyperpolarization), probably by repressing Ca^{2+} -dependent K^{+} currents; (3) MCU-mitochondrial Ca^{2+} uptake exerts a biphasic change on NAD(P)H autofluorescence, suggestive of a boost of both mitochondrial respiration and TCA metabolism.

Overall this is an interesting and solid study on important topics, though some conclusions are confirmatory. The experiments were well-executed, and the manuscript is well-written. I have only a few comments as the following.”

Comment 1:

“In the establishment of MCU as the principal route for mitochondrial Ca²⁺ uptake and ensuing signaling events within and beyond the organelle, the authors heavily relied on the single chemical agent, Ru360. Given the concern on promiscuous actions of this agent in cellular context, I am wondering whether they could adopt independent approaches, e.g., the use of MCU-knockout models?”

Response to Comment 1:

The reviewer refers to potential off-target actions of Ru360 and inquires about the feasibility of using a genetic knockdown approach to disrupt MCU function and validate our findings. Regarding Ru360, this drug has been demonstrated to be a selective MCU inhibitor. For example, Ru360 has no significant impact on various aspects of Ca²⁺ signalling, including endoplasmic reticulum Ca²⁺ release, plasma membrane Na⁺/Ca²⁺ exchanger, or voltage-gated Ca²⁺ current [25]. As the reviewer noted, this is critical to the interpretation of our results, and we detail this in the results section. In addition, we further examined the literature and found no indication that Ru360 influences non-MCU targets relevant to our study, such as voltage-gated Ca²⁺ channels, plasma membrane Ca²⁺ extrusion systems, ER Ca²⁺ release, or Ca²⁺-activated potassium channels. If the effects of Ru360 were mediated by its actions on targets other than the MCU, we might anticipate that this drug would affect Ca²⁺ signals and the slow afterhyperpolarization regardless of whether the stimulus paradigm was sufficient to elicit mitochondrial Ca²⁺ uptake by the MCU. However, as shown in Figure 6 and 7, the influences of Ru360 on Ca²⁺ signals and the afterhyperpolarization only occur during stimuli which evoke mitochondrial Ca²⁺ uptake (see Figure 3A, B). Therefore, we believe that the most parsimonious explanation for the effects of Ru360 are by its actions on the MCU. These points are emphasized in the manuscript (lines 273-280) because they are critical to the interpretation our results, as the reviewer indicated.

Regarding genetic knockdown of the MCU, this method would certainly be effective for confirming our findings and developing future studies on mitochondrial Ca²⁺ uptake and its role in brain function. There are various methods for interfering with endogenous MCU expression, such as the use of conditional MCU knockout mice [26], shRNA targeting of the MCU [27], or overexpression of a dominant negative MCU subunit [28]. These methods are viable options, but each would take several months to develop. For example, a floxed-MCU mouse is commercially available from the Jackson Laboratory (Strain #:029817) but getting to the experimental stage would take at least several months. This is because the experiment requires mouse strain cryorecovery, followed by breeding with an inducible Cre strain, knockdown induction, and validation. The delivery of shRNA or a dominant negative shRNA would require a custom AAV construct for *in vivo* delivery. In our experience, acquisition of a custom AAV can take several months prior to even starting experiments. Considering these time and cost obstacles and the noted selectivity of Ru360 for the MCU, we believe that the genetic knockdown approach is best suited for our future studies.

Comment 2:

“In addition, the authors could briefly discuss about a highly relevant work by Stoler et al, which appeared recently in eLife (Stoler et al., 2022).”

Response to Comment 2:

Thank you for highlighting this manuscript. Stoler *et al* (2022) recently performed experiments very similar to our work. Like our findings, their research demonstrates a frequency dependent coupling between activity and mitochondrial Ca^{2+} uptake^[29]. They also implicate mitochondrial Ca^{2+} uptake in the dendrites during spike-timing dependent plasticity, although a causal link was not examined^[29]. We think that this work complements our findings quite well and have referenced it in the discussion.

Comment 2 text changes:

Line 330-331: “This relationship was also described in another recent study on cortical pyramidal neurons in brain slices^[29]”

Line 440-444: “This dual role of mitochondrial Ca^{2+} uptake in the somatodendritic compartment could shape action potential output as well as synaptic integration and plasticity in the dendrites, where mitochondrial Ca^{2+} uptake was recently shown to occur during coincident pre- and post-synaptic activity^[29].”

References:

1. McCormick, D.A., et al., *Comparative electrophysiology of pyramidal and sparsely spiny stellate neurons of the neocortex*. J Neurophysiol, 1985. **54**(4): p. 782-806.
2. Thompson, J.K., M.R. Peterson, and R.D. Freeman, *Single-neuron activity and tissue oxygenation in the cerebral cortex*. Science, 2003. **299**(5609): p. 1070-2.
3. Brody, C.D., et al., *Timing and neural encoding of somatosensory parametric working memory in macaque prefrontal cortex*. Cereb Cortex, 2003. **13**(11): p. 1196-207.
4. Epsztein, J., M. Brecht, and A.K. Lee, *Intracellular determinants of hippocampal CA1 place and silent cell activity in a novel environment*. Neuron, 2011. **70**(1): p. 109-20.
5. Harvey, C.D., et al., *Intracellular dynamics of hippocampal place cells during virtual navigation*. Nature, 2009. **461**(7266): p. 941-6.
6. Klausberger, T., et al., *Brain-state- and cell-type-specific firing of hippocampal interneurons in vivo*. Nature, 2003. **421**(6925): p. 844-8.
7. Erisir, A., et al., *Function of specific K(+) channels in sustained high-frequency firing of fast-spiking neocortical interneurons*. J Neurophysiol, 1999. **82**(5): p. 2476-89.
8. Lin, Y., et al., *Brain activity regulates loose coupling between mitochondrial and cytosolic Ca(2+) transients*. Nat Commun, 2019. **10**(1): p. 5277.
9. Buzsaki, G., *Theta oscillations in the hippocampus*. Neuron, 2002. **33**(3): p. 325-40.
10. Nunez, A. and W. Buno, *The Theta Rhythm of the Hippocampus: From Neuronal and Circuit Mechanisms to Behavior*. Front Cell Neurosci, 2021. **15**: p. 649262.
11. Otto, T., et al., *Learning-related patterns of CA1 spike trains parallel stimulation parameters optimal for inducing hippocampal long-term potentiation*. Hippocampus, 1991. **1**(2): p. 181-92.
12. Degenetais, E., et al., *Electrophysiological properties of pyramidal neurons in the rat prefrontal cortex: an in vivo intracellular recording study*. Cereb Cortex, 2002. **12**(1): p. 1-16.
13. Funahashi, S., C.J. Bruce, and P.S. Goldman-Rakic, *Mnemonic coding of visual space in the monkey's dorsolateral prefrontal cortex*. J Neurophysiol, 1989. **61**(2): p. 331-49.
14. Kann, O., I.E. Papageorgiou, and A. Draguhn, *Highly energized inhibitory interneurons are a central element for information processing in cortical networks*. J Cereb Blood Flow Metab, 2014. **34**(8): p. 1270-82.
15. Spruston, N., et al., *Activity-dependent action potential invasion and calcium influx into hippocampal CA1 dendrites*. Science, 1995. **268**(5208): p. 297-300.
16. Callaway, J.C. and W.N. Ross, *Frequency-dependent propagation of sodium action potentials in dendrites of hippocampal CA1 pyramidal neurons*. J Neurophysiol, 1995. **74**(4): p. 1395-403.
17. Christie, B.R., et al., *Different Ca2+ channels in soma and dendrites of hippocampal pyramidal neurons mediate spike-induced Ca2+ influx*. J Neurophysiol, 1995. **73**(6): p. 2553-7.
18. Yellen, G., *Fueling thought: Management of glycolysis and oxidative phosphorylation in neuronal metabolism*. J Cell Biol, 2018. **217**(7): p. 2235-2246.
19. Diaz-Garcia, C.M., et al., *Neuronal Stimulation Triggers Neuronal Glycolysis and Not Lactate Uptake*. Cell Metab, 2017. **26**(2): p. 361-374 e4.
20. Diaz-Garcia, C.M., et al., *The distinct roles of calcium in rapid control of neuronal glycolysis and the tricarboxylic acid cycle*. Elife, 2021. **10**.
21. Duchen, M.R., *Ca(2+)-dependent changes in the mitochondrial energetics in single dissociated mouse sensory neurons*. Biochem J, 1992. **283** (Pt 1): p. 41-50.
22. Hayakawa, Y., et al., *Rapid Ca2+-dependent increase in oxygen consumption by mitochondria in single mammalian central neurons*. Cell Calcium, 2005. **37**(4): p. 359-70.
23. Chance, B. and H. Baltscheffsky, *Respiratory enzymes in oxidative phosphorylation. VII. Binding of intramitochondrial reduced pyridine nucleotide*. J Biol Chem, 1958. **233**(3): p. 736-9.

24. Shuttleworth, C.W., *Use of NAD(P)H and flavoprotein autofluorescence transients to probe neuron and astrocyte responses to synaptic activation*. *Neurochem Int*, 2010. **56**(3): p. 379-86.
25. Matlib, M.A., et al., *Oxygen-bridged dinuclear ruthenium amine complex specifically inhibits Ca²⁺ uptake into mitochondria in vitro and in situ in single cardiac myocytes*. *J Biol Chem*, 1998. **273**(17): p. 10223-31.
26. Kwong, J.Q., et al., *The Mitochondrial Calcium Uniporter Selectively Matches Metabolic Output to Acute Contractile Stress in the Heart*. *Cell Rep*, 2015. **12**(1): p. 15-22.
27. Qiu, J., et al., *Mitochondrial calcium uniporter Mcu controls excitotoxicity and is transcriptionally repressed by neuroprotective nuclear calcium signals*. *Nat Commun*, 2013. **4**: p. 2034.
28. Wu, Y., et al., *The mitochondrial uniporter controls fight or flight heart rate increases*. *Nat Commun*, 2015. **6**: p. 6081.
29. Stoler, O., et al., *Frequency- and spike-timing-dependent mitochondrial Ca²⁺ signaling regulates the metabolic rate and synaptic efficacy in cortical neurons*. *Elife*, 2022. **11**.

REVIEWERS' COMMENTS:

Reviewer #1 (Remarks to the Author):

The authors addressed all points and thoroughly revised the manuscript.

Reviewer #3 (Remarks to the Author):

The authors have addressed our concerns satisfactorily. We have no further comments.

REVIEWERS' COMMENTS:

Reviewer #1 (Remarks to the Author):

The authors addressed all points and thoroughly revised the manuscript.

Reviewer #3 (Remarks to the Author):

The authors have addressed our concerns satisfactorily. We have no further comments.

Reply:

We would like to thank the reviewers for committing their time to review our revised manuscript.